# FedGSNR: Accelerating Federated Learning on Non-IID Data via Maximum Gradient Signal to Noise Ratio

## Abstract

Federated learning (FL) allows participants jointly training a model without direct data sharing. In such a process, participants rather than the central server perform local updates of stochastic gradient descent (SGD) and the central server aggregates the gradients from the participants to update the global model. However, the non-iid training data in participants significantly impact global model convergence. Most of existing studies addressed this issue by utilizing variance reduction or regularization. However, these studies focusing on specific datasets lack theoretical guarantee for efficient model training. In this paper, we provide a novel perspective on the non-iid issue by optimizing Gradient Signal to Noise Ratio (GSNR) during model training. In each participant, we decompose local gradients calculated on the non-iid training data into the signal and noise components and then speed up the model convergence by maximizing GSNR. We prove that GSNR can be maximized by using the optimal number of local updates. Subsequently, we develop FedGSNR to compute the optimal number of local updates for each participant, which can be applied to existing gradient calculation algorithms to accelerate the global model convergence. Moreover, according to the positive correlation between GSNR and the quality of shared information, FedGSNR allows the server to accurately evaluate contributions of different participants (i.e., the quality of local datasets) by utilizing GSNR. Extensive experimental evaluations demonstrate that FedGSNR achieves on average a $1.69\times$ speedup with comparable accuracy.

## 1 Introduction

Federated learning (FL) McMahan et al. (2017) focuses on a practical scenario with multiple participants to collaboratively train a model without direct data sharing. Different from the typical centralized optimization problem, FL decomposes the optimization problem into several sub-optimization problems, and distributes them to different participants to be solved separately with the corresponding local datasets. Moreover, these local datasets often follow non-iid distributions in reality. During the training phase, each participant solves the sub-problem via stochastic gradient decent (SGD), and sends back the corresponding results for aggregation. One of the most popular FL algorithms is FedAvg McMahan et al. (2017), and it typically accelerates global model convergence through multiple local updates. Although it has shown great performance in many practical applications, there're still mysteries in this area, especially in non-iid cases, and many previous literatures Haddadpour & Mahdavi (2019); Khaled et al. (2020); Li et al. (2020b) make efforts to analyze the convergency or even to accelerate it.

One of the key challenges in FL is how a model can be well trained on non-iid data in different participants. On the one hand, such imbalance breaks the unbiased optimization procedure when we utilize multiple local updates. While on the other hand, due to the differences between different local datasets, non-iid information distribution makes it difficult to evaluate the contribution of different participants. The former slows down the FL-based model convergence, i.e., a key factor of the efficiency. The latter is associated with contribution evaluation involving malicious data tampering detection, contribution-based profit distribution, incentive mechanism design, etc. Especially in the current data-driven age Sim et al. (2020), contribution evaluation is particularly important. In addition

to separately pursuing the two goals of accelerating model convergence and improving contribution evaluation accuracy, these two goals even conflict with each other.

To solve above challenges, we propose a novel approach that speedups model training by maximizing Gradient Signal to Noise Ratio (GSNR). The intuitions behind the design are two-fold. First, there is always a global optimal solution no matter how the data is distributed. For each local dataset, we can obtain an optimal optimization direction, i.e., the global gradient. Second, based on the information theory, GSNR determines the channel capacity, i.e., Shannon's formula: $C = W \cdot \log(1 + \text{SNR})$, and a larger GSNR means we can get more information with identical communication rounds, which can accelerate the model convergence. Thus, we can decompose the local optimization direction (i.e., the local gradient) into mutually orthogonal signal vector and noise vector. We find that if we can obtain the global gradient, the signal vector is parallel to the global gradient, while the noise vector is orthogonal to it. Fig. 1 shows a typical example of orthogonal decomposition of two participants.

We prove that the number of local updates can control the GSNR value and we can maximize GSNR by computing the optimal number of local updates. To maximize GSNR, we utilize the gradients uploaded by the participants to estimate the global gradient, and propose a FedGSNR algorithm to compute the optimal number of local updates according to the estimated global gradient. Moreover, based on the GSNR perspective, we also develop a specific method to compute the GSNR for each dataset, which allow the server to evaluate each participant's contribution.

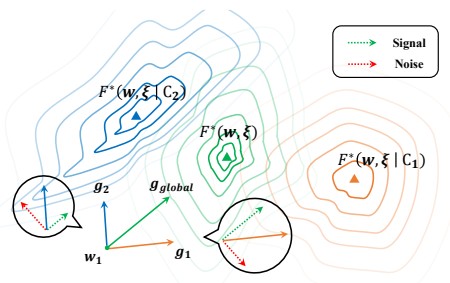

Figure 1: An example of Gradient Signal to Noise Ratio (GSNR). A local step can be decomposed into two components: signal and noise, the former parallels to the update with global data, and the latter orthogonal to it.

In addition to personalizing the number of local updates to optimize model convergence efficiency, the newly proposed GSNR strategy FedGSNR is orthogonal to existing methods, which mostly depends on modifying gradients calculating. Hence, FedGSNR can be combined with these methods so as to further improve them.

In summary, our contributions in this paper are as follows:

- We prove that the optimal local updates decides the maximal GSNR, which leads to faster and more stable convergence.
- We analyze existing FL algorithms with the perspective of GSNR. Moreover, based on the viewpoint of GSNR maximization, we propose an algorithm FedGSNR, which can be combined with most of current FL algorithms to calculate its optimal local updates.
- We derive a function $r(\boldsymbol{w})$ to calculate GSNR, which can be utilized to evaluate the local contributions of different participants.
- We confirm our theoretical results on CIFAR-10 and CIFAR-100 datasets, and experiments indicate that FedGSNR can achieve on average a $1.69\times$ speedup over its original when the information unevenly distributed among all participants, and $r(\boldsymbol{w})$ is a reasonable metric for local contributions.

**Related Work.** There has been a lot of literatures devoted to improving FL, including convergence Karimireddy et al. (2020); Li et al. (2020b); Wang et al. (2020a); Reddi et al. (2021) robustness Mohri et al. (2019); Fang et al. (2020); Li et al. (2021), and data privacy Melis et al. (2019); Zhu et al. (2019); Bagdasaryan et al. (2018). Regarding GSNR, Rainforth et al. (2018); Liu et al. (2020) try to analyze the generalization and variational bounds with such a concept. In this work, we focus on the relationship between GSNR and the optimal local updates in FL scenarios. To control the noise component (client drift), Karimireddy et al. (2020) proposes a specific gradient calculating method based on variance reduction. Li et al. (2020a) indicates that under non-iid FL conditions, a large number of local updates lead to divergence or instability. While Wang et al. (2020b) tries to stabilize the training procedure with a new average strategy. On the other hand, Wang et al. (2019) proposes a practical optimization problem with resources constraints, and it determines the number of local updates for each participant according to the corresponding constraints.

A similar work Khaled et al. (2020) derives an upper bound of local updates by total iterations $T$ and the number of participants $M$, which proposes a theoretical analysis of local updates. But, they treat each participant equally, and fail to propose a method to calculate the optimal number of local updates directly from the heterogeneous data. More discussion of related work can be found in Appendix A.

## 2 PRELIMINARY

### 2.1 FEDERATED AVERAGING (FEDAVG)

In this work, we consider the following federated optimization problem:

$$\min_{\boldsymbol{w}} \overline{F(\boldsymbol{w})} := \mathbb{E}_{\boldsymbol{\xi}}[F(\boldsymbol{w}, \boldsymbol{\xi})] = \mathbb{E}_C[\mathbb{E}_{\boldsymbol{\xi}}[F(\boldsymbol{w}, \boldsymbol{\xi})] \mid C] = \sum_{k=1}^{K} P(C = C_k) \cdot \mathbb{E}_{\boldsymbol{\xi}}[F(\boldsymbol{w}, \boldsymbol{\xi}|C_k)], \quad (1)$$

where $F(\boldsymbol{w}, \cdot)$ is a specified loss function with model $\boldsymbol{w}$, $K$ is the number of participants, and $P(C)$ is a discrete probability distribution correlated to the importance of different datasets. Usually, $P(C)$ is a uniform distribution or proportional to the local data quantity, and $\boldsymbol{\xi}|C_k$ is a random sample drawn from the dataset of $k$-th participant, i.e., $\boldsymbol{\xi}|C_k \sim p(\boldsymbol{x}|C_k)$.

Regarding traditional machine learning, the global dataset is gathered from all participants, and the goal is to minimize

$$F(\boldsymbol{w}) = \mathbb{E}_{\boldsymbol{\xi}}[F(\boldsymbol{w}, \boldsymbol{\xi})], \quad (2)$$

where $\boldsymbol{\xi}$ is a random sample of global dataset, i.e., the gathered data. However, in most cases, due to the privacy requests, we cannot gather data from different participants. Thus, we separate the target function as Eq. (1), and send the initial model $\boldsymbol{w}_1$ to each participant, then they do the optimization locally, and send back the corresponding results. We finally obtain the results by Eq. (1).

If each participant does only one step optimization, according to the property of conditional expectation, minimizing Eq. (1) is equivalent to minimizing Eq. (2). But this procedure puts a lot of pressure on communication, so researchers propose to do more local updates for efficiency. Hence, for the $k$-th participant, the optimization procedure of a typical round can be formalized as

$$\boldsymbol{w}_{i+1}^k \leftarrow \boldsymbol{w}_i^k - \eta \mathbb{E}_{\boldsymbol{\xi}|C_k}[\nabla_{\boldsymbol{w}} F(\boldsymbol{w}_i^k, \boldsymbol{\xi}|C_k)], \ i = 1, \cdots, n.$$

Then the central server aggregates local models $\boldsymbol{w}_{n+1}^1, \cdots, \boldsymbol{w}_{n+1}^K$ to update the global model by

$$\overline{\boldsymbol{w}} = \sum_{k=1}^{K} p_k \boldsymbol{w}_{n+1}^k, \quad (3)$$

where we denote $p_k$ for $P(C = C_k)$ for convenience.

### 2.2 WASSERSTEIN DISTANCE

Wasserstein distance (Villani (2009)) is a metric in probabilistic space inspired by the problem of optimal transport. It is a distance between probability distributions that takes geometric information into account. The general wasserstein distance is defined as

$$\boldsymbol{W}_p(\boldsymbol{\mu}, \boldsymbol{\nu}) = \inf_{\boldsymbol{\gamma} \in \boldsymbol{\Gamma}(\boldsymbol{\mu}, \boldsymbol{\nu})} \mathbb{E}_{(\boldsymbol{x}, \boldsymbol{y}) \sim \boldsymbol{\gamma}}[\|\boldsymbol{x} - \boldsymbol{y}\|_p],$$

which is difficult to find a closed form solution. However, if we chose 2-norm as the geometric measure and simplify the problem to Gaussian distribution, the distance has an analytic solution

$$d^2 = \|\boldsymbol{\mu}_1 - \boldsymbol{\mu}_2\|_2^2 + tr((\boldsymbol{\Sigma}_1^{\frac{1}{2}} - \boldsymbol{\Sigma}_2^{\frac{1}{2}})^2), \quad (4)$$

where we define $d := \boldsymbol{W}_2(\mathcal{N}(\boldsymbol{\mu}_1, \boldsymbol{\Sigma}_1), \mathcal{N}(\boldsymbol{\mu}_2, \boldsymbol{\Sigma}_2))$.

## 3 MAXIMIZE GSNR WITH OPTIMAL LOCAL UPDATES

In this section, we investigate the relationship between GSNR and the local updates, then we propose a method to calculate the optimal number of local updates. As Section 4 will introduce, GSNR can be

calculated by the ratio between the norm of global gradient, which is a constant for all participants, and the distance between global and local gradients. Therefore, to maximize GSNR is equivalent to minimize the distance between the distributions of global and local gradients. Furthermore, the minimal distance, i.e., the maximum GSNR, is decided by the optimal local updates.

On the other hand, as mentioned in Section 2.1, our target is to gather data so as to optimize Eq. (2) centralizedly. But in practice, due to some real restrictions, we can just optimize Eq. (1) distributedly. Thus, we treat the former procedure as an ideal optimization process. Based upon this idea, for accelerating convergence, our practical optimization problem also requires to minimize the distance between practical and ideal optimization path, which is determined by the corresponding gradients.

The distance between practical and ideal optimization path, which are denoted by the distributions of $p_{\overline{\boldsymbol{w}}}(\boldsymbol{w})$ and $p_{\boldsymbol{w}_g}(\boldsymbol{w})$ respectively[1], can be formalized as

$$D = \boldsymbol{W}_2(p_{\overline{\boldsymbol{w}}}(\boldsymbol{w}), p_{\boldsymbol{w}_g}(\boldsymbol{w})), \tag{5}$$

According to Eq. (3), due to the conditional independence of the data from different participants, the random vector $\overline{\boldsymbol{w}}$ is a convex combination of a set of independent random vectors, hence $p_{\overline{\boldsymbol{w}}}(\boldsymbol{w})$ is identified by the convolution of corresponding local distribution functions, and such a distribution contains all details of each participant. Intuitively, if a specific participant attempts to minimize Eq. (5), it has to gather all information from others, which violates the privacy requests. Then we need to derive an upper bound of Eq. (5) as

$$D = \boldsymbol{W}_2(p_{\overline{\boldsymbol{w}}}(\boldsymbol{w}), p_{\boldsymbol{w}_g}(\boldsymbol{w})) = \inf_{\boldsymbol{\gamma} \in \boldsymbol{\Gamma}(p_{\overline{\boldsymbol{w}}}, p_{\boldsymbol{w}_g})} \mathbb{E}_{(\boldsymbol{x}, \boldsymbol{y}) \sim \boldsymbol{\gamma}}[\|\boldsymbol{x} - \boldsymbol{y}\|_2]$$

$$\leq \sum_{k=1}^{K} p_k \inf_{\boldsymbol{\gamma_k} \in \boldsymbol{\Gamma}(p_{\boldsymbol{w}_{n+1}^k}, p_{\boldsymbol{w}_g})} \mathbb{E}_{(\boldsymbol{x}, \boldsymbol{y}) \sim \boldsymbol{\gamma_k}}[\|\boldsymbol{x} - \boldsymbol{y}\|_2], \tag{6}$$

where we split $\|\boldsymbol{x} - \boldsymbol{y}\|_2$ as $\|\sum_{k=1}^{K} p_k(\boldsymbol{x}_k - \boldsymbol{y})\|_2$, and each pair $(\boldsymbol{x}_k, \boldsymbol{y})$ is supported on $\boldsymbol{\Gamma}(p_{\boldsymbol{w}_{n+1}^k}, p_{\boldsymbol{w}_g})$. Then the inequality depends on triangle inequality and the fact that $\boldsymbol{w}_{n+1}^k$ is mutual independent. The upper bound is obvious, since by optimal transport, comprehensively considering all mounds is better than the sum of separate consideration.

By upper bound (6), the distance between practical and ideal optimization path is upper bounded. In other words, while minimizing upper bound (6), the target distance (5) is approximately minimized. Specifically, the target gap vanishes as the upper bound approaches to 0. Regarding Eq. (6), it is the sum of the distance between each independent local distribution $p_{\boldsymbol{w}_{n+1}^k}$ and the global distribution $p_{\boldsymbol{w}_g}$, then due to the rotating symmetry, we can minimize upper bound (6) separately for each participant. Hence, without loss of generality, we just consider a specific participant in federated learning in the rest of this paper.

Based on former analysis, for maximizing GSNR, each participant needs to greedily optimize its local gradient distribution to minimize the distance $\boldsymbol{W}_2(p_{\boldsymbol{w}_{n+1}^k}(\boldsymbol{w}), p_{\boldsymbol{w}_g}(\boldsymbol{w}))$. As the initial parameters, i.e., $\boldsymbol{w}_1^k$, for all participants are identical, the main target is to estimate the gradient distribution of different participants. Then we have the following assumption.

**Assumption 3.1. (Bounded variance)** The variance of stochastic gradients are uniformly bounded, i.e., $\mathbb{E}_{\boldsymbol{\xi}|C_i}\|\nabla_{\boldsymbol{w}}F(\boldsymbol{w}, \boldsymbol{\xi}|C_i) - \boldsymbol{\mu}_i\|^2 \leq \sigma^2$, $\forall i$, $\boldsymbol{w}$, where $\boldsymbol{\mu}_i := \mathbb{E}_{\boldsymbol{\xi}|C_i}[\nabla_{\boldsymbol{w}}F(\boldsymbol{w}, \boldsymbol{\xi}|C_i)]$.

Under such an assumption, the mini-batch stochastic gradient descent converges to joint normal distribution. The detailed proofs can be found in Appendix B.

**Lemma 3.2.** *With Assumption 3.1, let $\{\boldsymbol{\xi}_{i,b} \mid 1 \leq i \leq n; 1 \leq b \leq B\}$ be a set of iid samples of a specific dataset, $\boldsymbol{g} = (\boldsymbol{g}_1, \cdots, \boldsymbol{g}_n)$ be a finite dimensional gradient vector, where $\boldsymbol{g}_i = \frac{1}{B}\sum_{b=1}^{B} \nabla_{\boldsymbol{w}}F(\boldsymbol{w}_i, \boldsymbol{\xi}_{i,b})$, $i \in \{1, \cdots, n\}$, then $\sqrt{B}(\boldsymbol{g} - \mathbb{E}[\boldsymbol{g}])$ converges to multivariate normal distribution.*

**Remark 3.3.** Let $\boldsymbol{S} = \sqrt{B}(\boldsymbol{g} - \mathbb{E}[\boldsymbol{g}])$ and $\boldsymbol{Z} \sim \mathcal{N}(\boldsymbol{0}, \boldsymbol{\Sigma})$, where $\boldsymbol{\Sigma}$ is the covariance matrix of $\boldsymbol{S}$, then based on Berry–Esseen theorem, for all convex sets $\boldsymbol{U} \subseteq \mathbb{R}^d$, we have $|Pr(\boldsymbol{S} \in \boldsymbol{U}) - Pr(\boldsymbol{Z} \in \boldsymbol{U})| \leq C \frac{rank(\boldsymbol{\Sigma})^{1/4}}{B^{1/2}}$, where $C$ is a constant, which provides an upper bound of estimation error for Lemma 3.2.

---

[1] $\boldsymbol{w}_g$ is the ideal optimization path based on the global data distribution, i.e., the ideal distribution of data gathered from all participants, while $\overline{\boldsymbol{w}}$ is the corresponding practical path defined by Eq. (3).

Lemma 3.2 implies that, with mini-batch stochastic gradient descent, the sum of local updates converges to a Gaussian distribution, i.e., $\bar{\boldsymbol{g}} = \boldsymbol{w}_{n+1} - \boldsymbol{w}_1 = \sum_{i=1}^{n} \boldsymbol{g}_i = \mathbb{1}^T \boldsymbol{g}$, where $\bar{\boldsymbol{g}}$ is a linear transformation of a joint Gaussian vector. Thus, with a finite batch size $B$, it can be approximated by Gaussian distribution. Then we need to calculate the mean vector and the covariance matrix. For such a purpose, we have another assumption of smoothness.

**Assumption 3.4. (Smoothness)** The target function $F(\boldsymbol{w}, \cdot): R^m \to R$ is twice differentiable, and the expected matrix norm of hessian matrix $H(F(\boldsymbol{w}, \cdot))$ is bounded, i.e., $\mathbb{E}_{\boldsymbol{\xi}}\|H(F(\boldsymbol{w}, \boldsymbol{\xi}))\|^2 \leq L^2$, where $\boldsymbol{\xi}$ is randomly sampled from a specific dataset.

Note that Assumption 3.4 is weaker than L-smooth Assumption, since if a function $F(\boldsymbol{w}, \cdot)$ is L-smooth, it conforms to Assumption 3.4, but not vice versa.

Assumption 3.4 always holds for typical machine learning tasks, e.g., logistic regression, soft-max classification and so on. With these assumptions, we have the following lemma.

**Lemma 3.5.** *If Assumption 3.1 and 3.4 hold, let $\{\eta_r\}_{r=1}^{+\infty}$ be a sequence of real number such that $\lim_{r \to +\infty} \eta_r = 0$, and $\{\boldsymbol{\varepsilon}_r\}_{r=1}^{+\infty}$ be a sequence of random vectors, where $\boldsymbol{\varepsilon}_r = \hat{\boldsymbol{g}} - \bar{\boldsymbol{g}}$, and $\hat{\boldsymbol{g}} = \sum_{i=1}^{n} \nabla_{\boldsymbol{w}} F(\boldsymbol{w}_1, \boldsymbol{\xi}_i)$, $\bar{\boldsymbol{g}} = \sum_{i=1}^{n} \nabla_{\boldsymbol{w}} F(\boldsymbol{w}_i, \boldsymbol{\xi}_i)$ with $\boldsymbol{w}_i = \boldsymbol{w}_{i-1} - \eta_r \boldsymbol{g}_{i-1}, i \in \{2, \cdots, n\}$ respectively, then we have $\boldsymbol{\varepsilon}_r \xrightarrow{L} 0$, which implies $\hat{\boldsymbol{g}} \xrightarrow{L} \bar{\boldsymbol{g}}$.*

**Remark 3.6.** Based on the proof of Lemma 3.5 in appendix, the estimation error is $\mathbb{E}\|\bar{\boldsymbol{g}} - \hat{\boldsymbol{g}}\| \leq n(n-1)\eta_r LG$, which implies that if we consider learning rate decay[2], then

$$\forall \epsilon, \quad \lim_{r \to +\infty} Pr(\|\bar{\boldsymbol{g}} - \hat{\boldsymbol{g}}\| > \epsilon) = 0. \tag{7}$$

In a typical communication round $r$, the optimization process implies that $\boldsymbol{w}_{n+1} - \boldsymbol{w}_1 = \eta_r \bar{\boldsymbol{g}}$. While based on Eq. (7), we can use $\eta_r \hat{\boldsymbol{g}} = \eta_r \sum_{i=1}^{n} \nabla_{\boldsymbol{w}} F(\boldsymbol{w}_1, \boldsymbol{\xi}_i)$ to estimate $\eta_r \bar{\boldsymbol{g}}$. Moreover, if we multiply Eq. (7) by $\eta_r$, the estimation error becomes $\mathcal{O}((n\eta_r)^2 LG)$.

Specifically, since $\boldsymbol{w}_1$ is a constant vector and $\boldsymbol{\xi}_i$ is iid sampled from the dataset in different local steps, $\{\nabla_{\boldsymbol{w}} F(\boldsymbol{w}_1, \boldsymbol{\xi}_i) | i \in \{1, \cdots, n\}\}$ are also iid random vectors. Therefore, based on the sum of independent random variables, we have $\boldsymbol{\mu} = \mathbb{E}[\eta_r \hat{\boldsymbol{g}}] = n\eta_r \mathbb{E}_{\boldsymbol{\xi}}[F(\boldsymbol{w}_1, \boldsymbol{\xi})]$, and $\boldsymbol{\Sigma} = Cov(\eta_r \hat{\boldsymbol{g}}, \eta_r \hat{\boldsymbol{g}}) = n\eta_r^2 \boldsymbol{\Sigma}_1$.

Additionally, $\boldsymbol{\Sigma}$ is a second order term. To simplify the analysis, we need to convert coefficient $n$ to $n^2$. As mentioned before, $\boldsymbol{\Sigma}_1$ is the covariance matrix of a mean vector distribution, hence it depends on the batch size $B$. Let $B = \hat{B}/n$, then we obtain $\boldsymbol{\Sigma} = n^2\eta^2\boldsymbol{\Sigma}_1/\hat{B}$.

For convenience, in the rest of our work, we use $\boldsymbol{\mu}_*$ and $\boldsymbol{\Sigma}_*$ to denote the corresponding mean vector and covariance matrix of gradients estimated by a specific dataset. Similarly, as we can change the local batch size for simplifying computation, we ignore the differences between $B$ and $\hat{B}$.

According to Lemma 3.2 and 3.5, we can estimate the parameter distributions of one global step and $n$ local steps by $\mathcal{N}(\eta_r \boldsymbol{\mu}_g, \eta_r^2 \frac{\boldsymbol{\Sigma}_g}{B})$ and $\mathcal{N}(n\eta_r \boldsymbol{\mu}_l, n^2\eta_r^2 \frac{\boldsymbol{\Sigma}_l}{B})$ respectively. Then the optimal number of local updates to maximize GSNR is implied by following theorem.

**Theorem 3.7.** *The minimal Wasserstein distance between two multivariate Gaussian distribution denoted by $\mathcal{N}(\eta_r \boldsymbol{\mu}_g, \eta_r^2 \frac{\boldsymbol{\Sigma}_g}{B})$ and $\mathcal{N}(n\eta_r \boldsymbol{\mu}_l, n^2\eta_r^2 \frac{\boldsymbol{\Sigma}_l}{B})$ with variable $n$ is achieved when $n$ is*

$$n_1^{opt} = \max(0, \frac{\boldsymbol{\mu}_l^T \boldsymbol{\mu}_g + \frac{tr((\boldsymbol{\Sigma}_l \boldsymbol{\Sigma}_g)^{\frac{1}{2}})}{B}}{\|\boldsymbol{\mu}_l\|^2 + \frac{tr(\boldsymbol{\Sigma}_l)}{B}}),$$

*and the minimum distance is $(\Delta_1^{opt})^2 = \eta_r^2 \Delta^2$,*

$$\Delta^2 = \|\boldsymbol{\mu}_g\|^2 + \frac{tr(\boldsymbol{\Sigma}_g)}{B} - \frac{\left(\boldsymbol{\mu}_l^T \boldsymbol{\mu}_g + \frac{tr((\boldsymbol{\Sigma}_l \boldsymbol{\Sigma}_g)^{\frac{1}{2}})}{B}\right)^2}{\|\boldsymbol{\mu}_l\|^2 + \frac{tr(\boldsymbol{\Sigma}_l)}{B}}.$$

**Corollary 3.8.** *For $\mathcal{N}(m\eta_r \boldsymbol{\mu}_g, m^2\eta_r^2 \boldsymbol{\Sigma}_g)$ and $\mathcal{N}(n\eta_r \boldsymbol{\mu}_l, n^2\eta_r^2 \boldsymbol{\Sigma}_l)$, where $m$ is a constant, the optimal $n$ to minimize the Wasserstein distance is $n_m^{opt} = m \cdot n_1^{opt}$ and the minimal distance is $(\Delta_m^{opt})^2 = m^2 \cdot (\Delta_1^{opt})^2$.*

---

[2]For example, $\eta_r = \eta_0 \alpha^r$ refers to a widely used learning rate decay method with a decay rate $\alpha < 1$.

---

**Algorithm 1** Example of FedGSNR in conjunction with FedAvg

---

**Input:** initial model $\boldsymbol{w}_1$, learning rate $\eta_0$, sample size $B$, and chosen global steps $M$
**for** $r = 1$ **to** $R$ **do**
    Sample clients $\boldsymbol{S} \subseteq \{1, \cdots, K\}$
    **Server**: send $\boldsymbol{w}_1$ and $\eta_r$ to each client $i \in \boldsymbol{S}$
    **On each active client** $i$ **in parallel**: initialize local model $\boldsymbol{w}^i \leftarrow \boldsymbol{w}_1$, compute $\tilde{\boldsymbol{g}}_i$ and $diag(\tilde{\boldsymbol{\Sigma}}_i)$, and send them to the server
    **Server**: compute $n_{1,i}^{opt}$ with Theorem 3.7 for each client $i$, and send it to each client $i$
    **On each active client** $i$ **in parallel**:
        **for** $k = 1$ **to** $M \cdot n_{1,i}^{opt}$ **do**
            $\boldsymbol{w}^i \leftarrow \boldsymbol{w}^i - \eta_r \bar{\boldsymbol{g}}_i$
        **end for**
    **Server**: $\boldsymbol{w}_1 \leftarrow \sum_{i=1}^{|\tilde{\boldsymbol{S}}|} \tilde{p}_i \boldsymbol{w}^i$, where $\tilde{\boldsymbol{S}} = \{i | i \in \boldsymbol{S}, n_{1,i}^{opt} > 0\}$, and $\tilde{p}_i$ is the corresponding probability ratio, i.e., $p_i / \sum_{k \in \tilde{\boldsymbol{S}}} p_k$
**end for**

---

Based on Corollary 3.8, if $m$ is a constant, the minimum Wasserstein distance is achieved when $n = m * n_1^{opt}$, which is the optimal number of local updates for maximizing GSNR, leading to a maximal channel capacity for information communication.

To estimate the optimal local updates, we need to compute the mean vectors and covariance matrix for both local distribution and global distribution by the samples from each participant. In practice, we use sample mean vector and sample covariance matrix to estimate the parameters of local distribution, i.e., for participant $k$ with model $\boldsymbol{w}_1$, the corresponding statistics are $\tilde{\boldsymbol{g}}_k = \frac{1}{B} \sum_{b=1}^{B} \boldsymbol{g}_{k,b}$ and $\tilde{\boldsymbol{\Sigma}}_k = \frac{1}{B} \sum_{b=1}^{B} (\boldsymbol{g}_{k,b} - \tilde{\boldsymbol{g}}_k)(\boldsymbol{g}_{k,b} - \tilde{\boldsymbol{g}}_k)^T$, where $\boldsymbol{g}_{b,k} = \nabla_{\boldsymbol{w}} F(\boldsymbol{w}_1, \boldsymbol{\xi}_b | C_k)$. While for the server, based on the theorem of conditional random variables, the corresponding global statistics are

$$\tilde{\boldsymbol{g}} = \mathbb{E}_C[\tilde{\boldsymbol{g}}|C] = \sum_{k=1}^{K} p_k \tilde{\boldsymbol{g}}_k, \tag{8}$$

$$\tilde{\boldsymbol{\Sigma}} = \mathbb{E}_C[\tilde{\boldsymbol{\Sigma}}|C] + Cov_C(\tilde{\boldsymbol{g}}|C) = \sum_{k=1}^{K} p_k \tilde{\boldsymbol{\Sigma}}_k + \sum_{k=1}^{K} p_k [(\tilde{\boldsymbol{g}}_k - \tilde{\boldsymbol{g}})(\tilde{\boldsymbol{g}}_k - \tilde{\boldsymbol{g}})^T]. \tag{9}$$

In practice, as the covariance matrix increases the communication traffic and the calculation of matrix introduces lots of computation, we need to simplify the procedure. Specifically, in Theorem 3.7, we mainly need the trace of covariance matrix. Meanwhile, according to Balduzzi et al. (2017), we know that the covariance matrix of gradients is a sparse matrix and the estimate error can be scaled by the batch size $B$. Therefore, we can instead utilize the principal diagonal element of $\tilde{\boldsymbol{\Sigma}}_k$ for efficiency. Based on former analysis, we propose an algorithm FedGSNR to calculate the optimal number of local updates, and Algorithm 1 is a typical example of FedGSNR in conjunction with FedAvg[3].

**Partial participation.** In federated scenarios, the active participants are usually not 100%, so we cannot obtain perfect information of global gradient. However, we claim that FedGSNR can also adapt to this situation, because such imperfect information encourages us to transform Eq. (5) with triangle inequality to

$$\boldsymbol{W}_2(p_{\bar{\boldsymbol{w}}}(\boldsymbol{w}), p_{\boldsymbol{w}_g}(\boldsymbol{w})) \leq \boldsymbol{W}_2(p_{\bar{\boldsymbol{w}}}(\boldsymbol{w}), p_{\hat{\boldsymbol{w}}}(\boldsymbol{w})) + \boldsymbol{W}_2(p_{\hat{\boldsymbol{w}}}(\boldsymbol{w}), p_{\boldsymbol{w}_g}(\boldsymbol{w})),$$

where $\hat{\boldsymbol{w}}$ is the average parameters of active clients (i.e., a subset of total clients). With a specific client set $\boldsymbol{S}$, $\delta = \boldsymbol{W}_2(p_{\hat{\boldsymbol{w}}}(\boldsymbol{w}), p_{\boldsymbol{w}_g}(\boldsymbol{w}))$ is a constant, and $\boldsymbol{W}_2(p_{\bar{\boldsymbol{w}}}(\boldsymbol{w}), p_{\hat{\boldsymbol{w}}}(\boldsymbol{w}))$ can be bounded by inequality (6). Thus, with some tolerance $\delta$, we can similarly minimize Eq. (5) with the new upper bound, but the performance decreases as the ratio of active clients declines.

---

[3]Note that our proposed FedGSNR is a compatible method, and the referred FedAvg can also be replaced by other methods (e.g., FedProx).

**Convergence analysis.** FedGSNR is a convergent algorithm, since we just change the number of local updates. Moreover, we can transform the convergence analysis of FedGSNR to its original version by the inequality $1 \leq E_{min} \leq E_{i,r} \leq E_{max}$. We provide an example of convergence analysis for FedGSNR with FedAvg in Appendix E.

## 4 CALCULATE GSNR BY LOCAL GRADIENTS

In this section, we first analyze the optimal local updates as well as the optimal distance between the local gradient distribution and the global gradient distribution, and then derive a method to calculate GSNR by the optimal distance. However, due to the limited space, the detailed analysis between GSNR and the optimization procedure can be found in Appendix C.

Regarding the optimal number of local updates $n_1^{opt}$ and the corresponding optimal distance $(\Delta)^2$, let $L = \|\boldsymbol{\mu}_g\|^2 + \frac{tr(\boldsymbol{\Sigma}_g)}{B}$, $M = \boldsymbol{\mu}_l^T \boldsymbol{\mu}_g + \frac{tr((\boldsymbol{\Sigma}_l \boldsymbol{\Sigma}_g)^{\frac{1}{2}})}{B}$, and $N = \|\boldsymbol{\mu}_l\|^2 + \frac{tr(\boldsymbol{\Sigma}_l)}{B}$, we can rewrite $n_1^{opt} = \max(0, \frac{M}{N})$ and $(\Delta)^2 = L - \frac{M^2}{N}$.

Then we define a matrix as follows:

$$\boldsymbol{R}_* = \begin{pmatrix} u_*^1 & & & \vdots & \\ & u_*^2 & & \vdots & \\ & & \ddots & \vdots & \\ & & & u_*^t & \vdots \\ \hdashline & & & \vdots & \left(\frac{1}{B}\boldsymbol{\Sigma}_*\right)^{\frac{1}{2}} \end{pmatrix}$$

**Global Step**

$\eta^2 \left(\|\boldsymbol{\mu}_g\|^2 + \frac{tr(\boldsymbol{\Sigma}_g)}{B}\right)$

$\eta^2 \Delta^2$

$\theta$

**Local Steps**

Figure 2: An overview for GSNR: the GSNR can be calculated by the statistics of global gradient distribution and local gradient distribution.

where $\mu_*^i$ is the component of $\boldsymbol{\mu}_* = (\mu_*^1, \cdots, \mu_*^t)$.

On the one hand, $L = \|\boldsymbol{R}_g\|_F^2$ is correlated to the global distribution, which is a constant for all participants.[4] On the other hand, $N = \|\boldsymbol{R}_l\|_F^2$ depends on local distribution, thus it is a normalization coefficient. Hence, the two variables $n_1^{opt}$ and $(\Delta)^2$ mainly depend on the value of $M$, the inner product of two matrixes, i.e., $< \boldsymbol{R}_l, \boldsymbol{R}_g >_F$, which represents the similarity of them.

Based on former analysis, we derive a method to calculate GSNR as Definition 4.1.

**Definition 4.1. Gradient Signal to Noise Ratio (GSNR).** For a local dataset $\boldsymbol{D}_l$ and a global dataset $\boldsymbol{D}_g$, with a loss function $F(\boldsymbol{w}, \cdot)$, the GSNR is a function of $\boldsymbol{w}$ defined as

$$r(\boldsymbol{w}) = \max(0, \frac{< \boldsymbol{R}_l, \boldsymbol{R}_g >_F}{\sqrt{\|\boldsymbol{R}_l\|_F^2 \|\boldsymbol{R}_g\|_F^2 - < \boldsymbol{R}_l, \boldsymbol{R}_g >_F^2}}).$$

Specifically, Fig. 2 illustrates an example for Definition 4.1. In Fig. 2, we imagine a similar case in Euclid space. In this case, angle $\theta$ can be viewed as the similarity between global and local gradient vectors. In Euclid space, the minimum distance from a point to a line is the segment vertical to the line, thus the black dash line is orthogonal to local gradient vector. Hence, $\frac{\eta^2(\|\boldsymbol{\mu}_g\|^2 + tr(\boldsymbol{\Sigma}_g)/B)}{\eta^2 \Delta^2} = csc^2\theta$. As for GSNR, defined as the magnitude ratio between the parallel component and the orthogonal component, it can be viewed as the $\cot\theta$. Then according to trigonometric transformation, i.e., $\cot^2\theta = \csc^2\theta - 1$, we can obtain Definition 4.1.

## 5 EXPERIMENT

We run our experiments on the well known real world datasets CIFAR-10 and CIFAR-100 mentioned in Krizhevsky et al. (2009) to validate our design.

**Setup.** For non-iid settings, we utilize 3 methods for data partition. First, we follow the settings in Hsu et al. (2019) to generate non-iid data across different participants by Dirichlet distribution, where

---

[4]$\| \cdot \|_F$ and $< \cdot, \cdot >_F$ are Frobenius inner product and Frobenius norm respectively.

$\alpha$ is a parameter represents the level of non-iid. Second, we propose NonBalance and Pareto for imbalanced partition, which simulates the imbalanced distributed information in practical scenario. Due to the limited space, the details of different methods can be found in Appendix D. For all experiments, we use LeNet for CIFAR-10 and VGG-16 for CIFAR-100.

Table 1: Communication rounds to reach 0.5 accuracy and corresponding speedup[5] of FedGSNR on CIFAR10. We distributed the data among 30 clients, utilize batch size of 64, and set $E_{const} = 20$.

| Algorithms | $\alpha = 0.5$ | $\alpha = 0.1$ | Label 2 | NonBalance | Pareto |
|---|---|---|---|---|---|
| FedAvg | $170\,(1.0\times)$ | $355\,(1.0\times)$ | $470\,(1.0\times)$ | $210\,(1.0\times)$ | $550\,(1.0\times)$ |
| FedGSNR with FedAvg | $\mathbf{115\,(1.5\times)}$ | $\mathbf{285\,(1.3\times)}$ | $\mathbf{340\,(1.4\times)}$ | $\mathbf{155\,(1.4\times)}$ | $\mathbf{170\,(3.2\times)}$ |
| FedProx | $185\,(1.0\times)$ | $430\,(1.0\times)$ | $500\,(1.0\times)$ | $220\,(1.0\times)$ | $385\,(1.0\times)$ |
| FedGSNR with FedProx | $\mathbf{140\,(1.3\times)}$ | $\mathbf{300\,(1.4\times)}$ | $\mathbf{405\,(1.2\times)}$ | $\mathbf{180\,(1.2\times)}$ | $\mathbf{210\,(1.8\times)}$ |
| Scaffold | $385\,(1.0\times)$ | $770\,(1.0\times)$ | $870\,(1.0\times)$ | $390\,(1.0\times)$ | $>1K$ |
| FedGSNR with Scaffold | $\mathbf{140\,(2.7\times)}$ | $\mathbf{425\,(1.8\times)}$ | $\mathbf{770\,(1.1\times)}$ | $\mathbf{170\,(2.3\times)}$ | $>1K$ |

To ensure all methods are comparable, we need to set the total computation, i.e., local updates, to be equal. So in FedGSNR, we set the local updates for different participants to be $E_k = NE_{const}\frac{n_{1,k}^{opt}}{\sum_{i=1}^{K} n_{1,i}^{opt}}$, where $N$ and $E_{const}$ represent the active participants and the local updates of baseline algorithms respectively. Note that $E_k$ is a redistribution of local steps.

**The necessity of optimal local updates.** To understand the necessity of optimal local updates, we calculate the entropy of the local steps, i.e., $H = \sum_{k=1}^{K} p(C_k) \log p(C_k)$, where $p(C_k) = \frac{n_{1,k}^{opt}}{\sum_{i=1}^{K} n_{1,i}^{opt}}$, and the results are illustrated in Fig. 3. Specifically, the dashed blue line on the top is the uniform distribution of local updates, which is the maximum entropy distribution of discrete variables, and it represents the equal local updates among all participants. Moreover, when the degree of non-iid increases, the computation is allocated more concentrated, i.e., a

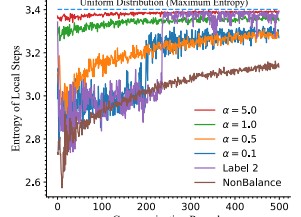
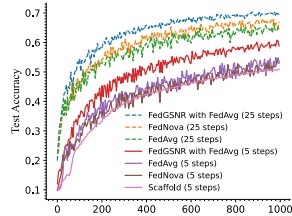

Figure 3: The entropy of local steps for different partition methods.

Figure 4: Test accuracy of different algorithms with different local steps.

smaller entropy. On the contrary, the entropy converges to its maximum when the distributed data is closer to iid. Additionally, the corresponding convergence rate is illustrated in Table 1, which indicates that the re-allocation of local updates based on FedGSNR accelerates the model convergence.

**The impatct on test accuracy.** Versus its original, FedGSNR achieves comparable test accuracy and even outperform its original when the non-iid degree increases. For example, in Pareto scenario, the accuracy of FedGSNR with FedProx achieves an increase of 6.43%. More detailed results can be found in Appendix D.2.

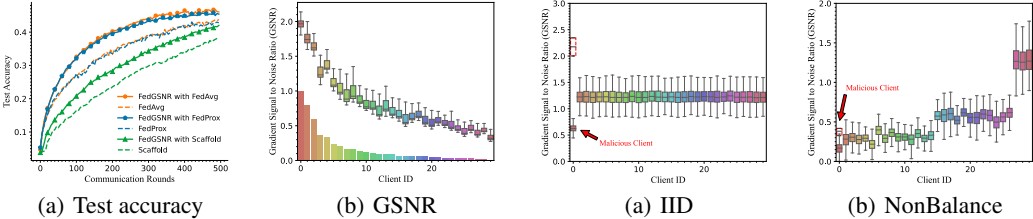

| (a) Test accuracy | (b) GSNR | (a) IID | (b) NonBalance |

Figure 5: An imbalanced scenario with Pareto partition on CIFAR-100 dataset.

Figure 6: The variation of GSNR when we change the labels of a specific participant.

**Model convergence of FedGSNR.** During experiments, we set $\mu = 0.01$ for FedProx, and compare the performance of different algorithms to its combination with FedGSNR. Then according to Table

---

[5]Speedup Karimireddy et al. (2020), i.e., $S = \frac{T_{old}}{T_{new}}$, measures the relative performance of two methods.

Table 2: Communication rounds to reach 0.5 test accuracy for classification on NonBalance CIFAR-10 of 100 participants as we vary the number of active clients.

|  | 10% | 20% | 100% |
|---|---|---|---|
| FedAvg | $210\,(1.0\times)$ | $140\,(1.0\times)$ | $100\,(1.0\times)$ |
| FedNova | $235\,(0.9\times)$ | $140\,(1.0\times)$ | $80\,(1.2\times)$ |
| Scaffold | $230\,(0.9\times)$ | - | - |
| FedGSNR with FedAvg | $\mathbf{150\,(1.4\times)}$ | $\mathbf{80\,(1.7\times)}$ | $\mathbf{55\,(1.8\times)}$ |

1, FedGSNR with different algorithms achieve faster convergence versus its original, and it reaches a $1.69\times$ speedup on average with comparable accuracy. The corresponding accuracy can be found in Appendix D.2. Besides, Fig. 4 illustrates the accuracy when we set different number of $E_{const}$, and FedGSNR with FedAvg converges faster and achieves better accuracy with different local steps (Scaffold fails to work when we set $E_{const} = 25$). Moreover, in practice, Pareto's Law is a common principle, which means a small number of participants possess a large number of information. Fig. 5(a) indicates that FedGSNR with different algorithms converge faster and reaches comparable accuracy. Meanwhile, the GSNR of different participants are resemble to their label distribution (the histogram on the bottom of Fig. 5(b)), which demonstrates GSNR can distinguish the information quality between different local datasets. Furthermore, Table 2 indicates that the growth of active clients speeds up the convergence of different algorithms. Particularly, FedGSNR gains more benefit from global information as its speedup increases from $1.4\times$ to $1.8\times$ when active clients grows.

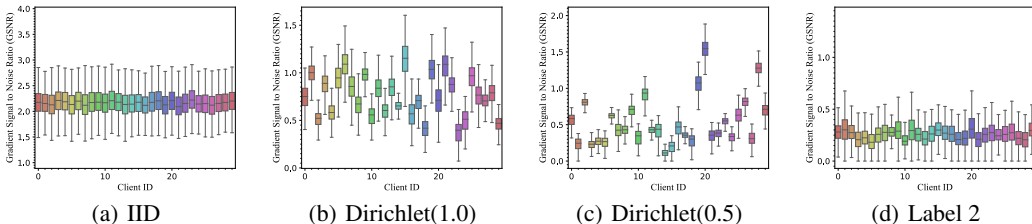

| (a) IID | (b) Dirichlet(1.0) | (c) Dirichlet(0.5) | (d) Label 2 |
|---|---|---|---|

Figure 7: The GSNR of different clients. We observe that GSNR is larger and almost the same among all participants when data is iid distributed, then it gets smaller and heterogeneous as the non-iid level grows. Finally, when data partition method is Label 2, GSNR is small but similar to each other again, which probably indicates that data is distributed with some symmetries in regard to the information.

**Evaluate local contributions with GSNR.** Fig. 7 displays the variation of GSNR when we utilize Dirichlet method with different $\alpha$. And the results demonstrate that when the level of non-iid grows, GSNR of different clients vary dramatically, which represents the contributions of different clients are different. Moreover, combined GSNR with the results in Table 1, the model convergence is faster than its opponents when we considerate such differences. Further more, to investigate the performance of data evaluation, we change the labels $l$ of client 0 to be $(l + k) \, mod \, 10$. So that the client provides a label flipping attack as Hitaj et al. (2017). Fig. 6 illustrates the changes of GSNR when we change the labels, and the red dashed line box represents the original GSNR when the labels are unchanged. Specifically, we observe that the GSNR dramatically decreases when we make a malicious change to the labels. Additionally, we instead change the data points to be sampled from a uniform distribution, and observe a similar phenomenon. For both of malicious changes, we observe FedGSNR is more robust. Due to the limited space, we send these experiments to Appendix D.

## 6 CONCLUSION

In this paper, we have investigated the FL problem via a new perspective, i.e., GSNR. Our theoretical analysis indicates that under non-iid scenarios, the local updates can be decomposed into signal and noise components, and we can maximize GSNR with the optimal local updates. Based on theoretical analysis, we further propose an algorithm FedGSNR to calculate the optimal local updates for different FL algorithms, which achieves faster global model convergence. Additionally, we derive a method to calculate GSNR directly from the local datasets, which can be utilized to evaluate the local contributions of different participants. Finally, extensive experimental results demonstrate the beneficial effect of optimizing FL from the new perspective of GSNR, and also open up a promising new direction for follow-up research.

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

## A    RELATED WORK

**Gradient Diversity.** Gradient diversity is a key ingredient of federated learning, which captures the differences between the datasets possessed by different participants. Yin et al. (2018) employs gradient diversity to investigate the relationship between batch size and the convergence rate in parallel SGD. Yu et al. (2019) analyzes why periodical model averaging is suitable for deep learning, and provides a deep understanding of model averaging. Haddadpour et al. (2019) tries to mitigate gradient diversity through sharing a small batch of data among all participants, but it also introduces a higher privacy risk. Acar et al. (2021) introduces a dynamic regularization term to resolve the problem of gradient divergence. However, most of previous literatures try to solve gradient diversity through gradient calculating, such as gradient prediction, regularization, personalized target function and so on. However, the influence of local updates gains less attention. In this paper, we propose a new perspective to analyze the optimization procedure by Gradient Signal to Noise Ratio, it reduces the required communication rounds via an elaborate configuration of local updates and propose a method to evaluate the contributions of different participants.

**Personalization in federated learning.** Another important problem in federated learning is personalization. Formally, personalization transform the optimization problem from global distribution $p(\boldsymbol{\xi})$ to a specific local distribution $p(\boldsymbol{\xi} \mid C_i)$ on client $i$, it scarifies the global performance in order to gain more benefit in local scenario. Kulkarni et al. (2020) reviews the investigations of personalization, and the situations are divided into three categories: device heterogeneity, data heterogeneity and model heterogeneity, while the last one is the motivation of personalization. Mansour et al. (2020) proposes three methods to achieve personalization, these methods try to balance the model performance on global data distribution as well as local data distribution. Sim et al. (2019) proposes a method to optimize global model and local model separately in order to make local model more personalized. Jiang et al. (2019) proposes three objectives to make personalization easily. However, personalization is an important topic in federated learning since different participants confront different problems, but if we greedily utilize global information for personalization, there is likely to appear Prisoner's Dilemma, the collective benefit for all participants is not optimal and therefore, the profit for each participant can probably be futher improved. Hence, cooperation is also an important problem, and a better goal of personalization is to search optimum on conditional data distribution $p(\boldsymbol{\xi} \mid C_i)$ combined with cooperation.

## B    PROOFS OF LEMMA AND THEOREM

### B.1    PROOF OF LEMMA 3.2

*Proof.* For any constant $n$, gradient vector $\boldsymbol{g}$ can be rewritten as

$$\boldsymbol{g} = \frac{1}{B}\sum_{b=1}^{B}(\nabla_{\boldsymbol{w}}F(\boldsymbol{w}_1, \boldsymbol{\xi}_{1,b}), \cdots, \nabla_{\boldsymbol{w}}F(\boldsymbol{w}_n, \boldsymbol{\xi}_{n,b})),$$

let $\tilde{\boldsymbol{g}}_b = (\nabla_{\boldsymbol{w}}F(\boldsymbol{w}_1, \boldsymbol{\xi}_{1,b})), \cdots, \nabla_{\boldsymbol{w}}F(\boldsymbol{w}_n, \boldsymbol{\xi}_{n,b})))$, we further have

$$\boldsymbol{g} = \frac{1}{B}\sum_{b=1}^{B}\tilde{\boldsymbol{g}}_b,$$

then with Assumption 3.1 and $n$ is a constant, $\tilde{\boldsymbol{g}}_b$ is subject to some complex distribution with bounded covariance matrix. As $\boldsymbol{\xi}_{i,b}$ is iid sampled from a specific dataset, $\boldsymbol{g}$ is the mean vector of $\tilde{\boldsymbol{g}}_1, \cdots, \tilde{\boldsymbol{g}}_B$, which are iid random vectors.
Therefore, based on the classical Central Limit Theory, with $B$ growing large, $\sqrt{B}(\boldsymbol{g} - \mathbb{E}[\boldsymbol{g}])$ converges to $\mathcal{N}(\boldsymbol{0}, \boldsymbol{\Sigma})$ in distribution, where $\boldsymbol{\Sigma}$ is the covariance matrix of $\boldsymbol{g}$. $\qquad\square$

### B.2    PROOF OF LEMMA 3.5

*Proof.* First, we prove $\lim_{r\to+\infty}\mathbb{E}\|\boldsymbol{\varepsilon}_r\| = 0$.

Regarding the gradient $\boldsymbol{g}_i$, $i \in \{1, \cdots, n\}$, due to the smoothness of $\nabla_{\boldsymbol{w}} F(\boldsymbol{w}_i, \boldsymbol{\xi})$, we can expand $\boldsymbol{g}_i$ based on Lagrange's mean value theorem as

$$\boldsymbol{g}_i = \nabla_{\boldsymbol{w}} F(\boldsymbol{w}_i, \boldsymbol{\xi}_i) = \nabla_{\boldsymbol{w}} F(\boldsymbol{w}_1, \boldsymbol{\xi}_i) + H(F(\tilde{\boldsymbol{w}}_i, \boldsymbol{\xi}_i))(\boldsymbol{w}_i - \boldsymbol{w}_1) = \boldsymbol{g}_{1,i} + H(F(\tilde{\boldsymbol{w}}_i, \boldsymbol{\xi}_i))(\boldsymbol{w}_i - \boldsymbol{w}_1),$$
(10)

where $\tilde{\boldsymbol{w}} := \lambda \boldsymbol{w}_i + (1 - \lambda) \boldsymbol{w}_1$, $\lambda \in [0, 1]$, and $\boldsymbol{g}_{1,i}$ represents the gradient at $\boldsymbol{w}_1$ with sample $\boldsymbol{\xi}_i$, which is an unbiased estimator of $\boldsymbol{g}_1$. Note that when $i \neq j$, $\boldsymbol{g}_{1,i}$ is independent of $\boldsymbol{g}_{1,j}$. Then

$$\mathbb{E}\|\boldsymbol{g}_i - \boldsymbol{g}_{1,i}\| = \mathbb{E}\|H(F(\tilde{\boldsymbol{w}}_i, \boldsymbol{\xi}_i))(\boldsymbol{w}_i - \boldsymbol{w}_1)\| \overset{(a)}{\leq} \mathbb{E}\|H(F(\tilde{\boldsymbol{w}}_i, \boldsymbol{\xi}_i))\|\|(\boldsymbol{w}_i - \boldsymbol{w}_1)\| \quad (11)$$

$$\overset{(b)}{\leq} \sqrt{\mathbb{E}\|H(F(\tilde{\boldsymbol{w}}_i, \boldsymbol{\xi}_i))\|^2 \mathbb{E}\|\boldsymbol{w}_i - \boldsymbol{w}_1\|^2} \overset{(c)}{\leq} L \cdot \sqrt{\mathbb{E}\|\eta_r \sum_{j=1}^{i-1} \boldsymbol{g}_j\|^2}$$

$$\overset{(d)}{\leq} L^2 \cdot \eta_r \sqrt{(i-1) \sum_{j=1}^{i-1} \mathbb{E}\|\boldsymbol{g}_j\|^2} \overset{(e)}{\leq} (i-1)\eta_r L G,$$

where $(a)$ follows from sub-multiplicative property of matrix norm, $(b)$ is based on Cauchy–Schwarz inequality, $(c)$ is an immediate consequence of Assumption 3.4 and the local optimization process, $(d)$ comes from the fact $\|\sum_{i=1}^n \boldsymbol{a}_i\|^2 \leq n \sum_{i=1}^n \|\boldsymbol{a}_i\|^2$, and $(e)$ is based on Assumption 3.1, where $G^2 := \sigma^2 + \mu^2$, $\mu = \max(\{\|\boldsymbol{\mu}_i\|\}_{i \in \{1, \cdots, n\}})$. Hence,

$$\mathbb{E}\|\boldsymbol{\varepsilon}_r\| = \mathbb{E}\|\hat{\boldsymbol{g}} - \bar{\boldsymbol{g}}\| = \mathbb{E}\|\sum_{i=1}^n (\boldsymbol{g}_i - \boldsymbol{g}_{1,i})\| \leq \sum_1^n \mathbb{E}\|\boldsymbol{g}_i - \boldsymbol{g}_{1,i}\|$$

$$\leq (\sum_{i=1}^n (i-1))\eta_r L G = \frac{n(n-1)}{2}\eta_r L G$$

where the first inequality follows from the triangle inequality, and the second inequality is based on Eq. (11).

As $n$ represents the number of local steps, which is a constant, $\mathbb{E}\|\boldsymbol{\varepsilon}_r\|$ is upper bounded by $\eta_r \cdot M$, where $M$ is a bounded value. Therefore,

$$0 \leq \lim_{r \to +\infty} \mathbb{E}\|\boldsymbol{\varepsilon}_r\| \leq \lim_{r \to +\infty} \eta_r \cdot M = 0. \quad (12)$$

Formula (12) implies $\boldsymbol{\varepsilon}_r$ converge to 0 in mean, i.e., $\boldsymbol{\varepsilon}_r \overset{L}{\to} 0$, which immediately completes the proof. $\square$

### B.3 PROOF OF THEOREM 3.7

*Proof.* According to Eq. (4), to minimize the distance between $\mathcal{N}(\eta_r \boldsymbol{\mu}_g, \eta_r^2 \frac{\boldsymbol{\Sigma}_g}{B})$ and $\mathcal{N}(n\eta_r \boldsymbol{\mu}_l, n^2 \eta_r^2 \frac{\boldsymbol{\Sigma}_l}{B})$, we can build an optimization problem as

$$\min_n \quad d^2 = \|\eta_r \boldsymbol{\mu}_g - n\eta_r \boldsymbol{\mu}_l\|^2 + tr(\boldsymbol{M}^2) \quad (13)$$

$$\text{s.t.} \quad \boldsymbol{M} = \left(\frac{\eta_r^2 \boldsymbol{\Sigma}_g}{B}\right)^{\frac{1}{2}} - \left(\frac{n^2 \eta_r^2 \boldsymbol{\Sigma}_l}{B}\right)^{\frac{1}{2}}$$

$$n \geq 0.$$

Note that Eq. (13) is a quadratic function of $n$, which immediately completes the proof. $\square$

### B.4 Proof of Corollary 3.8

*Proof.* In this case, we change the distribution $\mathcal{N}(\eta_r \boldsymbol{\mu}_g, \eta_r^2 \frac{\boldsymbol{\Sigma}_g}{B})$ to $\mathcal{N}(m\eta_r \boldsymbol{\mu}_g, m\eta_r^2 \frac{\boldsymbol{\Sigma}_g}{B})$, and reformulate problem (13) as

$$\min_n \quad d^2 = m^2(\|\eta_r \boldsymbol{\mu}_g - \frac{n}{m}\eta_r \boldsymbol{\mu}_l\|^2 + tr(\boldsymbol{M}^2))$$

$$\text{s.t.} \quad \boldsymbol{M} = \left(\frac{\eta_r^2 \boldsymbol{\Sigma}_g}{B}\right)^{\frac{1}{2}} - \left(\frac{\left(\frac{n}{m}\right)^2 \eta_r^2 \boldsymbol{\Sigma}_l}{B}\right)^{\frac{1}{2}}$$

$$n \geq 0,$$

let $\tilde{d} = \frac{d}{m}$ and $\tilde{n} = \frac{n}{m}$, then the new problem reduces to problem (13), which concludes the proof immediately. $\square$

## C  Detail Analysis of GSNR

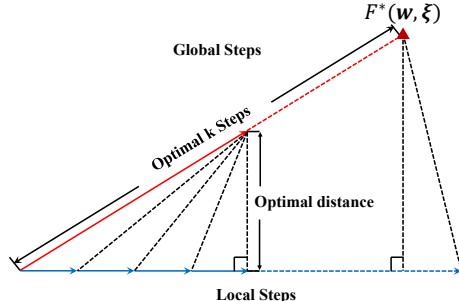

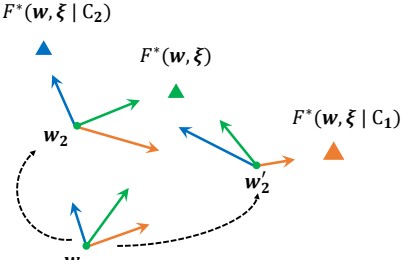

Figure 8: A similar case in Euclid space: as the ideal optimization path is discrete, there is an optimal distance between global updates and multiple local updates. Further more, when global updates converge to the optimum, the local updates converge to the nearest point to the optimum.

Figure 9: A representative scenario of GSNR: $r(\boldsymbol{w})$ is a random variable with regard to $\boldsymbol{w}$. If we get closer to the optimum of $C_2$, GSNR of $C_1$ will increase and the GSNR of $C_2$ will decrease. While if we get closer to the optimum of $C_1$, the situation changes in the opposite way.

According to Fig. 8, the stochastic gradient descent algorithm converges to the $\epsilon$-neighborhood of optimum after a constant steps (usually more than $\mathcal{O}(\frac{1}{\epsilon})$ Nesterov (1998)). For convenience, we refer such a constant as the optimal $m_{opt}$. In practice, as Fig. 8 indicates, with determined differences between global distribution and local distribution, i.e., maximal GSNR is a constant during a specific round, set the target global steps as $m_{opt}$ is an optimal choice. However, a large number of $m$ leads to a large error of gradient estimation, which is determined by $O(\eta^2 n^2)$. Hence, in practice, in order to decide $m$, we need to trade off between the estimation error and the corresponding convergence rate.

Then we focus on Definition 4.1, the function to calculate GSNR. Specifically, Cauchy-Schwarz inequality implies that $\Delta^2 \geq 0$, and the equality holds when local distribution is the same as the global distribution, i.e., the data is iid distributed among all participants. With $\Delta^2$ decreases, which implies local data distribution approaches the global data distribution, $n_1^{opt}$ gradually increases. Moreover, when $\Delta^2$ reaches its minimum 0, $n_1^{opt}$ attains its maximum value 1. Based on the analysis, we can conclude that the more similarity between local dataset and the global dataset, i.e., the larger GSNR the local dataset achieves, the more local updates we need for optimization procedure, which is heuristically experimented in Li et al. (2020b).

To calculate GSNR, we derive $r(\boldsymbol{w})$ as Definition 4.1, and $r(\boldsymbol{w})$ is positive related to $n_1^{opt}$. On the one hand, when $n_1^{opt} = 0$, from its definition, we know that $< \boldsymbol{R}_l, \boldsymbol{R}_g >_F = 0$, hence the optimal distance $\Delta^2$ achieves its maximum $\|\boldsymbol{R}_g\|_F^2$, and $r(\boldsymbol{w})$ attains its minimum 0. On the other hand, when local distribution is the same as the global distribution, i.e., $\Delta^2 = 0$, $r(\boldsymbol{w}) \to +\infty$, we treat this

scenario as a noiseless optimization procedure, and the data is iid distributed among each participant. Therefore, $r(\boldsymbol{w}) \in (0, +\infty)$.

With former analysis, we know $r(\boldsymbol{w}) \in (0, +\infty)$. On the one hand, as the data is iid distributed among all participants, i.e., GSNR goes to $+\infty$, the distributed optimization is a noiseless procedure, which means the local updates is unbiased. On the other hand, when GSNR is 0, which means $< \boldsymbol{R}_l, \boldsymbol{R}_g >_F \leq 0$, the angle between local gradient and the global gradient is greater than $90°$. In other words, for current optimization, local data distribution is independent of global data distribution, thus for global optimization, it is no better than a random guess, then its signal component will be set to 0, which leads GSNR to be 0.

As for the relationship between GSNR and the parameters $\boldsymbol{w}$, Fig. 9 displays a representative scenario. Due to the randomness of SGD, the new parameters after $\boldsymbol{w}_1$ with another aggregation can be either $\boldsymbol{w}_2$ or $\boldsymbol{w}_2'$. On the one hand, if the parameters is $\boldsymbol{w}_2$, which means we get closer to the optimum of client $C_2$, there are different changes of the GSNR for different clients: for $C_1$, the GSNR increases, while for $C_2$, the vector is almost orthogonal to global optimization vector, which implies its GSNR is closer to 0. On the other hand, if the parameters is $\boldsymbol{w}_2'$, which is closer to optimum of $C_1$, the phenomenon is slightly different: the GSNR of $C_1$ decreases and the GSNR of $C_2$ increases. Hence, during the training process, $r(\boldsymbol{w})$ is a random variable correlated to the random process $\boldsymbol{w}$, and if we tend to use GSNR to evaluate the contribution of different participants, we need to observe its statistics, i.e., mean or median.

# D  DETAILS OF EXPERIMENTS

## D.1  DIFFERENT METHODS OF DATA PARTITION

**Dirichlet partition.** we follow the settings in Hsu et al. (2019) to generate non-iid data across different participants by Dirichlet distribution. Specifically, the prior distribution is set to be Uniform, and then the parameter $\alpha$ represents the level of concentration. With $\alpha \to +\infty$, the data distributions of all participants tend to be identical, hence the data is iid distributed among all clients. While $\alpha \to 0$, each participant only possesses data chosen from just one class, i.e., one label for each participant. As for **Label 2.**, it is a specific partition method in Hsu et al. (2019), and each client owns the data sampled from 2 classes.

**NonBalance partition.** For NonBalance partition, we tend to simulate the practical scenario of imbalanced information distribution. Specifically, We divide all participants into three categories: abundant information, medium information and less information, which represent the clients possess data chosen from different number of labels. First, for clients with abundant information, we random chose data from all labels, and the number of them is $10\%$ of total clients. Second, for the clients with medium information, we random chose $50\%$ classes for each client, then distribute data randomly according to their chosen labels, and the ratio of them is $40\%$. Finally, for the clients with less information, the number of labels reduces to $20\%$, and the ratio of them raises to $50\%$.

**Pareto partition.** In practice, Pareto distribution is a common scenario. It represents the long tailed distribution of practical scenario such as the degree of nodes in complex network, the distribution of social wealth, the distribution of followers in social network, etc. Hence, we design Pareto partition to simulate the so called Two-Eight distribution in practice. First, we sample $N$ points from Pareto distribution,

$$p(x) = \begin{cases} \frac{k \cdot x_{min}^k}{x^{k+1}}, & \text{if } x \geq x_{min} \\ 0, & \text{otherwise.} \end{cases}$$

Where $N$ represents the number of clients. Denote the corresponding samples as $\boldsymbol{X} = \{x_i\}_{i=1}^N$, and we normalize $x_i$ with $\tilde{x}_i = x_i / \max(\boldsymbol{X})$ to guarantee all data distributed in [0, 1]. Finally, we use $\tilde{x}_i$ as the ratio of classes possessed by different clients for random sampling, and set the minimum number of labels among all clients to be 1.

## D.2  ADDITIONAL EXPERIMENTS

Fig. 10 displays the change of GSNR and corresponding test accuracy when we apply different malicious change to a client. Interestingly, compare Fig. 10(a) and 10(b) with Fig. 10(c) and

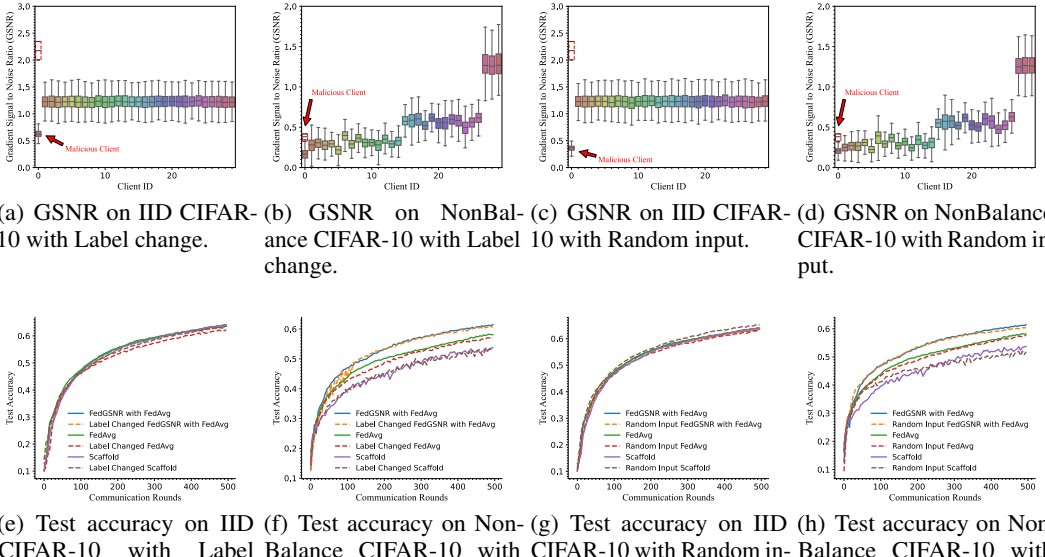

(a) GSNR on IID CIFAR-10 with Label change.

(b) GSNR on NonBalance CIFAR-10 with Label change.

(c) GSNR on IID CIFAR-10 with Random input.

(d) GSNR on NonBalance CIFAR-10 with Random input.

(e) Test accuracy on IID CIFAR-10 with Label change.

(f) Test accuracy on Non-Balance CIFAR-10 with Label change.

(g) Test accuracy on IID CIFAR-10 with Random input.

(h) Test accuracy on Non-Balance CIFAR-10 with Random input.

Figure 10: We make different malicious changes to client $0$ in different partition methods. In (a) and (b), we change the labels $l$ of client $0$ to $l + 3$. While in (c) and (d), the labels remain unchanged, and we change the input data to be a uniform distribution $U(-1, 1)$. (e)-(h) are the corresponding test accuracy of different scenarios.

10(d), we can discover that the decrease of GSNR for random input is larger than label changing. Specifically, it is consistent with our intuition, since malicious attack of label changing contains more information than random input. Simply put, after label changing, the model still gets the information that such data belong to a same class. For example, if we change all labels of 'cat' to 'dog', we still know the data of 'cat' belongs to a same class, though we call them 'dog'. On the contrary, the change of random input cannot provide this information.

Table 3: Best test accuracy on CIFAR-10. We distributed the data among 30 clients, utilize batch size of 64, and set local steps $E_{const} = 20$ for different algorithms.

| Algorithms | $\alpha = 0.5$ | $\alpha = 0.1$ | Label 2 | NonBalance | Pareto |
|---|---|---|---|---|---|
| FedAvg | **67.06%** | 57.44% | **58.25%** | 65.48% | 57.23% |
| FedGSNR with FedAvg | 66.23% | **61.41%** | 57.69% | **66.42%** | **63.38%** |
| FedProx | 63.14% | 57.80% | **58.84%** | 63.87% | 56.96% |
| FedGSNR with FedProx | **63.5%** | **59.39%** | 58.39% | **68.17%** | **63.39%** |
| Scaffold | 62.05% | 53.98% | 50.38% | 62.79% | **40.73%** |
| FedGSNR with Scaffold | **63.53%** | **58.95%** | **55.48%** | **64.96%** | 39.83% |

Table 3 displays the corresponding test accuracy of different algorithms aforementioned in Table 1, and it indicates that FedGSNR not only converges faster, but also achieves a comparable accuracy than its opponents. As for the accuracy drop in Table 3, it's possibly because FedGSNR is a gradient-based algorithm, if the basic method introduces gradient estimation (i.e., Scaffold), the performance of FedGSNR will be correlated to the precision of such an estimation, and a relatively low precision leads to the corresponding accuracy drop. Meanwhile, as we have illustrated in Fig. 7, the method of Label 2 distributes the data with some symmetries in regard to the information, i.e., the GSNR of each participant are similar to each other, then it's naturally compatible to identical local updates, hence the test accuracy between FedGSNR and its opponents are close to each other.

# E   AN EXAMPLE OF CONVERGENCE ANALYSIS

As defined in Sec. 4, we have

$$
\boldsymbol{R}_* =
\begin{pmatrix}
u_*^1 & & & & \vdots \\
& u_*^2 & & & \vdots \\
& & \ddots & & \vdots \\
& & & u_*^t & \vdots \\
\hdashline
& & & & \left(\frac{1}{B}\boldsymbol{\Sigma}_*\right)^{\frac{1}{2}}
\end{pmatrix}
$$

where $\boldsymbol{\mu}_*$ and $\boldsymbol{\Sigma}_*$ are the corresponding mean vector and covariance matrix calculated by the samples sampled from $\boldsymbol{D}_*$ respectively, and $\mu_*^i$ is the component of $\boldsymbol{\mu}_* = (\mu_*^1, \cdots, \mu_*^t)$. Then we have following lemmas.

**Lemma E.1.** $\|\boldsymbol{R}_*\|^2 \geq \frac{1}{B}\mathbb{E}_{\boldsymbol{\xi}}[\|\boldsymbol{g}_*\|^2]$, where $B \geq 1$.

*Proof.* First, we have

$$
\boldsymbol{\mu}_*^T\boldsymbol{\mu}_* = \mathbb{E}_{\boldsymbol{\xi}}[\boldsymbol{g}_*]^T\mathbb{E}_{\boldsymbol{\xi}}[\boldsymbol{g}_*] = tr(\mathbb{E}_{\boldsymbol{\xi}}[\boldsymbol{g}_*]\mathbb{E}_{\boldsymbol{\xi}}[\boldsymbol{g}_*]^T),
$$

Note that $B \geq 1$, hence

$$
\|\boldsymbol{R}_*\|^2 = \boldsymbol{\mu}_*^T\boldsymbol{\mu}_* + \frac{tr(\boldsymbol{\Sigma}_*)}{B} \geq \frac{\boldsymbol{\mu}_*^T\boldsymbol{\mu}_* + tr(\boldsymbol{\Sigma}_*)}{B} = \frac{tr(\mathbb{E}_{\boldsymbol{\xi}}[\boldsymbol{g}_*]\mathbb{E}_{\boldsymbol{\xi}}[\boldsymbol{g}_*]^T + \boldsymbol{\Sigma}_*)}{B}
$$
$$
= \frac{\mathbb{E}_{\boldsymbol{\xi}}[tr(\boldsymbol{g}_*\boldsymbol{g}_*^T)]}{B} = \frac{1}{B}\mathbb{E}_{\boldsymbol{\xi}}[\|\boldsymbol{g}_*\|^2],
$$

which concludes the proof. $\qquad\square$

**Lemma E.2.** *If Assumption 3.1 holds, then $\|\boldsymbol{R}_*\|^2 \leq G^2$, where $G^2$ is the upper bound of $\mathbb{E}_{\boldsymbol{\xi}}[\|\boldsymbol{g}_*\|^2]$.*

*Proof.* Similarly,

$$
\|\boldsymbol{R}_*\|^2 = \boldsymbol{\mu}_*^T\boldsymbol{\mu}_* + \frac{tr(\boldsymbol{\Sigma}_*)}{B} \leq \boldsymbol{\mu}_*^T\boldsymbol{\mu}_* + tr(\boldsymbol{\Sigma}_*) = tr(\mathbb{E}_{\boldsymbol{\xi}}[\boldsymbol{g}_*]\mathbb{E}_{\boldsymbol{\xi}}[\boldsymbol{g}_*]^T + \boldsymbol{\Sigma}_*)
$$
$$
= \mathbb{E}_{\boldsymbol{\xi}}[tr(\boldsymbol{g}_*\boldsymbol{g}_*^T)] = \mathbb{E}_{\boldsymbol{\xi}}[\|\boldsymbol{g}_*\|^2] \leq G^2,
$$

which concludes the proof. $\qquad\square$

Then we analyze the convergence of FedGSNR with FedAvg based on the proofs of Li et al. (2020b). For the analysis, we make additional assumptions.

**Assumption E.3.** The functions $F_k$ are all L-smooth: for all $\boldsymbol{w}$ and $\boldsymbol{v}$, $F_k(\boldsymbol{v}) \leq F_k(\boldsymbol{w}) + (\boldsymbol{v} - \boldsymbol{w})^T\nabla F_k(\boldsymbol{w}) + \frac{L}{2}\|\boldsymbol{v} - \boldsymbol{w}\|_2^2$.

**Assumption E.4.** The functions $F_k$ are all u-strongly convex: for all $\boldsymbol{w}$ and $\boldsymbol{v}$, $F_k(\boldsymbol{v}) \geq F_k(\boldsymbol{w}) + (\boldsymbol{v} - \boldsymbol{w})^T\nabla F_k(\boldsymbol{w}) + \frac{\mu}{2}\|\boldsymbol{v} - \boldsymbol{w}\|_2^2$.

**Assumption E.5.** $\forall k, k \in \{1, \cdots, K\}, \mathbb{E}_{\boldsymbol{\xi}}[\|\nabla F_k(\boldsymbol{w})\|^2] \leq G^2$.

Where $F_k(\boldsymbol{w})$ is short for $F(\boldsymbol{w}; \boldsymbol{\xi}|C_k)$.

The Lemma 1 and 2 in Li et al. (2020b) implies the bound for one step SGD and the bound for the variance of gradients, as they are independent of the number of local steps, we can use them directly.

**Lemma E.6.** *(Results of one step SGD). Assume Assumption E.3 and E.4 hold. If $\eta_t \leq \frac{1}{4L}$, we have*

$$
\mathbb{E}\|\boldsymbol{v}_{t+1} - \boldsymbol{w}_*\|^2 \leq (1 - \eta_t\mu)\mathbb{E}\|\bar{\boldsymbol{w}}_t - \boldsymbol{w}_*\|^2 + \eta_t^2\mathbb{E}\|\boldsymbol{g}_t - \bar{\boldsymbol{g}}_t\|^2 + 6L\eta_t^2\Gamma + 2\mathbb{E}[\sum_{k=1}^{K} p_k\|\bar{\boldsymbol{w}}_t - \boldsymbol{w}_k^t\|],
$$

*where $\Gamma = F^* - \sum_{k=1}^{K} p_k F_k^* \geq 0$.*

**Lemma E.7.** *(Bounding the variance). Assume Assumption 3.1 holds, then we have*

$$\mathbb{E}\|\boldsymbol{g}_t - \bar{\boldsymbol{g}}_t\| \leq \sum_{k=1}^{K} p_k^2 \sigma_k^2 \leq \sigma^2.$$

We focus on Lemma 3 in Li et al. (2020b), which is related to the number of local steps, and we prove that when we apply our method to decide the number of local steps, the lemma still holds.

**Lemma E.8.** *(Bounding the divergence of $\boldsymbol{w}_t^k$ when we use FedGSNR to decide the number of local updates). Assume the assumptions hold, then we have*

$$\mathbb{E}[\sum_{k=1}^{K} p_k \|\bar{\boldsymbol{w}}_t - \boldsymbol{w}_k^t\|^2] \leq \frac{4\eta_t^2 E_{const}^2 BG^2 L}{\mu}.$$

*where $\bar{\boldsymbol{w}}_t = p_k \boldsymbol{w}_k^t$.*

*Proof.* Since based on our strategy, the local steps is individually decided by $E_{k,r} = n_{1,k,r}^{opt} * E_{const}$, where $E_{const}$ is a constant. Hence we define the local optimization process as

$$\boldsymbol{w}_k^{t+1} = \begin{cases} \boldsymbol{w}_k^t - \nabla_{\boldsymbol{w}} F(\boldsymbol{w}_k^t), \ 0 \leq t - t_0 < E_{k,r} \\ \boldsymbol{w}_k^t, \ E_{k,r} \leq t - t_0 < E_{max}, \end{cases} \tag{14}$$

where $\boldsymbol{w}_{t_0}^k = \bar{\boldsymbol{w}}_{t_0}$ represents the aggregation step, and without loss of generality, $t_0$ is the initial time of communication round $r$, $E_{max} = \max\{E_{k,r} \mid 1 \leq k \leq K, 1 \leq r \leq R\}$. Then the situation becomes an identical number of local steps, and we prove that the upper bound is independent of $E_{max}$. We use the fact $t - t_0 < E_{max}$, where $t_0$ represents the last aggregation step before $t$, $\eta_t$ is non-increasing and $\eta_{t_0} \leq 2\eta_t$ for all $t - t_0 \leq E_{max}$, we have

$$\begin{aligned}
\mathbb{E}[\sum_{k=1}^{K} p_k \|\bar{\boldsymbol{w}}_t - \boldsymbol{w}_k^t\|^2] &= \mathbb{E}[\sum_{k=1}^{K} p_k \|(\boldsymbol{w}_k^t - \bar{\boldsymbol{w}}_{t_0}) - (\bar{\boldsymbol{w}}_t - \bar{\boldsymbol{w}}_{t_0})\|^2] \\
&\overset{(1)}{\leq} \mathbb{E}[\sum_{k=1}^{K} p_k \|(\boldsymbol{w}_k^t - \bar{\boldsymbol{w}}_{t_0})\|^2] \\
&\overset{(2)}{=} \sum_{k=1}^{K} p_k \mathbb{E}[\|\sum_{s=t_0}^{\min\{t,t_0+E_{k,r}-1\}} \eta_s^2 \nabla_{\boldsymbol{w}} F(\boldsymbol{w}_k^s)\|^2] \\
&\overset{(3)}{\leq} \sum_{k=1}^{K} p_k \mathbb{E}[\sum_{s=t_0}^{t_0+E_{k,r}-1} E_{k,r} \eta_s^2 \|\nabla_{\boldsymbol{w}} F(\boldsymbol{w}_k^s)\|^2] \\
&\overset{(4)}{\leq} \sum_{k=1}^{K} p_k \eta_{t_0}^2 \mathbb{E}[\sum_{s=t_0}^{t_0+E_{k,r}-1} E_{const} \frac{\|\boldsymbol{R}_g^{t_0}\|}{\|\boldsymbol{R}_k^{t_0}\|} \|\nabla_{\boldsymbol{w}} F(\boldsymbol{w}_k^s)\|^2] \\
&\overset{(5)}{\leq} \sum_{k=1}^{K} p_k \eta_{t_0}^2 E_{const} \mathbb{E}[\underbrace{\sum_{s=t_0}^{t_0+E_{k,r}-1} \frac{\sqrt{B}G}{\sqrt{\mathbb{E}[\|F(\boldsymbol{w}_k^{t_0})\|^2]}} \|\nabla_{\boldsymbol{w}} F(\boldsymbol{w}_k^s)\|^2}_{A_1}],
\end{aligned}$$

where inequality (1) depends on $\bar{\boldsymbol{w}}_t = p_k \boldsymbol{w}_k^t$ and the fact $\|a + b\|^2 \leq 2(\|a\|^2 + \|b\|^2)$. Inequality (2) is based on the local optimization process Eq. (14). Inequality (3) is a consequence of $\|\sum_{i=1}^{n} a_i\|^2 \leq n \sum_{i=1}^{n} \|a_i\|^2$. Inequality (4) depends on the Cauchy-Schwarz inequality, i.e., $n_1^{opt} \leq \frac{\|\boldsymbol{R}_g\|\|\boldsymbol{R}_l\|}{\|\boldsymbol{R}_l\|^2} = \frac{\|\boldsymbol{R}_g\|}{\|\boldsymbol{R}_l\|}$, and inequality (5) is an immediate consequence of Lemma E.1 and E.2. Then based on the L-Smoothness, we have

$$\|\nabla_{\boldsymbol{w}} F(\boldsymbol{w}_k^s)\|^2 \leq 2L(F(\boldsymbol{w}_k^s) - F(\boldsymbol{w}^*)),$$

similarly, according to the $\mu$-strong convexity, we have

$$\|\nabla_{\boldsymbol{w}} F(\boldsymbol{w}_k^s)\|^2 \geq 2\mu(F(\boldsymbol{w}_k^s) - F(\boldsymbol{w}^*)).$$

Hence, we can further bound $A_1$ as

$$
\begin{aligned}
A_1 &\leq \sum_{k=1}^{K} p_k \eta_{t_0}^2 E_{const} \sum_{s=t_0}^{t_0+E_{k,r}-1} \frac{\sqrt{B}G}{\sqrt{2\mu(F(\boldsymbol{w}_k^{t_0}) - F(\boldsymbol{w}^*))}} (2L(F(\boldsymbol{w}_k^s) - F(\boldsymbol{w}^*))) \\
&\leq \sum_{k=1}^{K} p_k \eta_{t_0}^2 E_{const} \sum_{s=t_0}^{t_0+E_{k,r}-1} \frac{\sqrt{B}G}{\sqrt{2\mu(F(\boldsymbol{w}_k^{t_0}) - F(\boldsymbol{w}^*))}} (2L(F(\boldsymbol{w}_k^{t_0}) - F(\boldsymbol{w}^*))) \\
&\leq \sum_{k=1}^{K} p_k \eta_{t_0}^2 E_{const}^2 \frac{BG^2}{2\mu(F(\boldsymbol{w}_k^{t_0}) - F(\boldsymbol{w}^*))} (2L(F(\boldsymbol{w}_k^{t_0}) - F(\boldsymbol{w}^*))) \\
&\leq \frac{4\eta_t^2 E_{const}^2 BG^2 L}{\mu},
\end{aligned}
$$

$\square$

where the second inequality depends on $F(\boldsymbol{w}^*) = \min_{\boldsymbol{w}} F(\boldsymbol{w})$, and the fact that $F(\boldsymbol{w}_k^{t+1}) \leq F(\boldsymbol{w}_k^t)$, $\forall t \in \{t_0, \cdots, t_0 + E_{k,r} - 1\}$ (i.e., the process is a non-increasing sequence). The third inequality depends on the upper bound of $E_{k,r}$, and the last inequality is a consequence of $\eta_{t_0} \leq 2\eta_t$.

Hence, we have

$$
\mathbb{E}[\sum_{k=1}^{K} p_k \|\bar{\boldsymbol{w}}_t - \boldsymbol{w}_k^t\|^2] \leq \frac{4\eta_t^2 E_{const}^2 BG^2 L}{\mu},
$$

which is related to a constant $E_{const}$.

Then we can prove the Thm. 1 in Li et al. (2020b) by the substitution for the upper bound of $\mathbb{E}[\sum_{k=1}^{K} p_k \|\bar{\boldsymbol{w}}_t - \boldsymbol{w}_k^t\|^2]$.

**Theorem E.9.** *Let the assumptions hold, and $L$, $\mu$, $G$, $\sigma$ defined therein, if we choose $\kappa = \frac{L}{\mu}$, $\gamma \geq \max\{8\kappa - 1, E_{max}\}$ and the learning rate $\eta_t = \frac{2}{\mu(\gamma+t)}$. Then FedGSNR with FedAvg satisfies*

$$
\mathbb{E}[F(\boldsymbol{w}_T)] - F^* \leq \frac{2\kappa}{\gamma + T}(\frac{M}{\mu} + 2L\|\boldsymbol{w}_0 - \boldsymbol{w}^*\|), \tag{15}
$$

*where*

$$
M = \sigma^2 + 6L\Gamma + \frac{8E_{const}^2 BG^2 L}{\mu}.
$$

