# OpenReview forum: "FedGSNR: Accelerating Federated Learning on Non-IID Data via Maximum Gradient Signal to Noise Ratio"
_ICLR.cc/2023/Conference — Submitted to ICLR 2023_

### Official Review · Reviewer_Dx1S · 2022-10-25

**Confidence:** 2
**Clarity, Quality, Novelty And Reproducibility:** In Strength And Weaknesses
**Correctness:** 2
**Technical Novelty And Significance:** 2
**Empirical Novelty And Significance:** 2
**Recommendation:** 6

**Strength And Weaknesses:**

This paper speedups model training by maximizing GSNR in FL and provides the optimal local updates from a theoretical viewpoint. However, I'm not an expert in FL and have a heavy review load. Due to my limited time, I can't be more positive at the current stage. I encourage the authors pay more attention on other reviewers' responses as I will make my confidence the lowest.

Though I'm not an expert in FL but I find the theoretical results in Sec 3 are interesting. Thus, I will only give some comments and ask some questions about this part. My questions may be highly biased due to I didn't go through all details in the paper. I would be glad if the authors could point out my misunderstandings.

1. About the upper bound Eqn (6). Why is the target distance (5) approximately minimized when the upper bound is minimized? There might be a gap. It would be better if the authors provide a simple example to verify its possibility in the current setting.
2. I'm confused by Lemma 3.2 and Lemma 3.4. It seems Lemma 3.4 is to calculate the asymptotic mean and variance by initial weight $w_1$. And the step size goes infinity. But Lemma 3.2 let the minibatch $B$ goes infinity. Could the author tell me more about how to connect Lemma 3.2 and Lemma 3.4 with different asymptotic parameters ($B$ and $r$)?



**Summary Of The Paper:**

In Strength And Weaknesses

**Summary Of The Review:**

In Strength And Weaknesses

---

> ### Author Response · Authors · 2022-11-13
> **Author Reply**
>
> We thank the reviewer for the comments and reviews.
>
> **Comment 1:** About the upper bound Eq. (6). Why is the target distance (5) approximately minimized when the upper bound is minimized? There might be a gap. It would be better if the authors provide a simple example to verify its possibility in the current setting.
>
> **Reply:** We agree with the reviewer that there may be a gap between the upper bound and the target distance. However, if the minimum of the upper bound is $0$, the upper bound becomes tighter when the optimization process gets closer to the minimum. Otherwise, the algorithm can provide an upper bound guarantee for the target distance under rigorous privacy restrictions.\
> Moreover, the reason why we minimize the upper bound Eq. (6) instead of minimizing the target distance Eq. (5) is that we cannot get detailed information of local data in order to minimize target distance Eq. (5) due to the restrictions for privacy guarantee.
>
> **Comment 2:** I'm confused by Lemma 3.2 and Lemma 3.4. It seems Lemma 3.4 is to calculate the asymptotic mean and variance by initial weight w1. And the step size goes infinity. But Lemma 3.2 let the minibatch B goes infinity. Could the author tell me more about how to connect Lemma 3.2 and Lemma 3.4 with different asymptotic parameters (B and r)?
>
> **Reply:** The relationships between Lemma 3.2 and 3.4 with asymptotic parameters depend on the estimation error. Accrdong to Lemma 3.2, the estimation error is proportional to $\frac{\sqrt[4]{rank(Σ)}}{\sqrt{B}}$, which means a larger batch size leads to a better approximation. As for Lemma 3.4, the estimation error is $\mathcal{O}((n η_r)^2 LG)$, which indicates that if we consider learning rate decay, i.e., $η_r \to 0$, the estimation error decreases when the optimization process gets closer to the optimum. We have added Remark 3.3 and Remark 3.6 to discuss the estimation errors in different lemmas.

---

### Official Review · Reviewer_8mk1 · 2022-10-26

**Confidence:** 2
**Correctness:** 3
**Technical Novelty And Significance:** 2
**Empirical Novelty And Significance:** 2
**Recommendation:** 3

**Clarity, Quality, Novelty And Reproducibility:**

Clarity and quality:

I enjoyed the overall story of the paper, but when I tried to check the details I was bothered by some clarity issues.

Is Lemma 3.4 saying that running gradient descent is equivalent to running gradient descent without updating the gradient? This approximation seems too crude to give a good approximation of the optimal number of local steps. Besides, I would have expected the optimal number of local steps to be just one (since the communication constraint is not encoded in the GSNR)?

In Eq 5, p_w are not defined. It is not clear what this equation means.

Th 3.5. is the main theorem, but its statement is really confusing. The optimal number of local updates is not mathematically defined. What is the criterion to be optimized (I needed to check the proof to understand)? Are there constraints? Same for the optimal distance (here it is also not clear what "distance" refers to). I suggest putting the proof of Th 3.5 in the main text.

Finally, these optimal numbers are obtained after making several approximations: CLT, Lemma 3.4. This is not recalled, nor discussed.

The GSNR appears only in page 7, after it is used to compute the optimal number of local steps. Wy this choice? This is confusing. Moreover, in the current form, Section 2.2 seems useless because it is not related to the rest of the paper (only implicitly for the proof of Th 3.5), am I correct?


Experiments show the flexibility of the approach to evaluate local contribution, this is an interesting feature of the approach. They also check the test accuracy, justify the necessity of optimal local steps and obtain a convergence speedup. There are some experiments for justification of local steps, but I am not sure to understand what the outcome of this experiment is. However, they do show model convergence speedup.

Novelty:

The idea of monitoring the deviation between local and global training is original. I believe that it could give good results in practice.


**Strength And Weaknesses:**

Strength

- The approach to compute the optimal number of local updates is original and flexible, can be applied to virtually any local optimization algorithm

-The method allows for a different number of local updates for each client, and allows for measuring the contribution of each client

Weaknesses

-GSNR is computed on the basis of several approximations. We don't know how far is the estimated optimal number of local updates from the true optimal number of local updates. In general, the fact that the approach is based on several approximations is really not discussed

-The goal of the GSNR approach is to accelerate the training. Acceleration is only shown experimentally

**Summary Of The Paper:**

In Federated Learning with heterogeneous data, the gradient of each client is pointing in a different direction when running a local algorithm s.t. local SGD. Since each client performs several local steps, the algorithm deviates from the theoretically better algorithm that would follow the averaged direction of the gradients.

This paper proposes a technique to reduce this deviation effect and speedup the training algorithm. Based on the Gradient Signal to Noise Ratio, they compute the "optimal" number of local steps to reduce the deviation effect. In particular, the number of local steps is not the same for every client.

The approach relies on making some gaussian approximation for the local training and the global training, computing the Wasserstein distance for the deviation between both and computing the number of local steps to minimize the deviation.

**Summary Of The Review:**

The idea is original but not well executed.

---

> ### Author Response · Authors · 2022-11-13
> **Author Reply**
>
> **Comment 5:** Thm 3.5. is the main theorem, but its statement is really confusing. The optimal number of local updates is not mathematically defined. What is the criterion to be optimized (I needed to check the proof to understand)? Are there constraints? Same for the optimal distance (here it is also not clear what "distance" refers to). I suggest putting the proof of Thm 3.5 in the main text.
>
> **Reply:** Theorem 3.5 states that the minimal wasserstein distance of two Gaussian distributions, which are denoted by $\mathcal{N}(η_r\mu_g,  \frac{η_r^2 \Sigma_g}{B})$ and $\mathcal{N}( n η_r\mu_l,  \frac{n^2 η_r^2 \Sigma_l}{B})$, with regard to variable $n$ is achieved when $n=n^{opt}_1$. The optimal distance is the minimum wasserstein distance of aforementioned two Gaussian distributions when $n=n^{opt}_1$.\
> We have rephased Theorem 3.5 in the revision.
>
> **Comment 6:** The GSNR appears only in page 7, after it is used to compute the optimal number of local steps. Why this choice? This is confusing. Moreover, in the current form, Section 2.2 seems useless because it is not related to the rest of the paper (only implicitly for the proof of Thm 3.5), am I correct?
>
> **Reply:** We present the definition of GSNR in page 7 because we can only define the method of calculating GSNR after we get the optimal wasserstein distance between global and local gradient distributions.\
> Additionally, Section 2.2 is related to our analysis, since the wasserstein distance is a distance for different distributions, which considers the geometrical information that is crucial for our analysis. We clarifies this issue in the revision.
>
> **Comment 7:** Experiments show the flexibility of the approach to evaluate local contribution, this is an interesting feature of the approach. They also check the test accuracy, justify the necessity of optimal local steps and obtain a convergence speedup. There are some experiments for justification of local steps, but I am not sure to understand what the outcome of this experiment is. However, they do show model convergence speedup.
>
> **Reply:** The experiment for justification of local steps is displayed in Fig.3, which shows the relationship between the degree of non-iid and the optimal allocation of local steps. To our best knowledge, it is the first study to show such a relationship.\
> Specifically, the entropy of the distribution of local steps measures the allocation imbalance (i.e., a smaller entropy of local steps means a more imbalanced allocation). Fig. 3 illustrates that when the data distributes with higher degree of non-iid, the allocation of local steps is more concentrated with a smaller entropy. Moreover, according to Table 1, the imbalanced allocation indeed accelerates the training, which means the optimal number of local updates is important for non-iid scenarios.

---

> > ### Comment · Reviewer_8mk1 · 2022-11-14
> > **Thanks for the answers**
> >
> > I understand that acceleration should be achieved by maximizing the information sent to the server at each iteration. But, is this a mathematical statement?
> >
> > Moreover, I am still bothered by the fact that the main result relies on several approximations. I understand that they are necessary to obtain the main result, but I actually don't know what it means to maximize the approximation. What if the approximation is too crude? The result of the procedure could be meaningless.
> >
> > Finally, I think that the exposition could be improved. The main message is not easy to understand. For example, in the current version, Th 3.7 seems to be an easy math statement about the Wasserstein distance between Gaussians. I know that this is not the case, and that the Gaussians appearing here are related to the original SNR problem, but this is the kind of issues that the reader has to deal with.

---

> > > ### Author Response · Authors · 2022-11-18
> > > **Thanks for your response**
> > >
> > > First of all, thanks for your kindly response.
> > >
> > > **Comment 1:** I understand that acceleration should be achieved by maximizing the information sent to the server at each iteration. But, is this a mathematical statement?
> > >
> > > **Reply:** The rigorous mathematical relationship between the channel capacity and the convergence is really an important and difficult problem, which is correlated to the interpretability of information flow for Deep Learning. Moreover, the mathematical relationship between the information flow and the training of Deep Learning has been raised as a grand challenge [1] (please see https://arxiv.org/pdf/2103.11251.pdf for details). Our paper focuses on developing a new perspective to interpret Federated Learning through information theory and improve the performance by modeling the process as a problem of information transmission.
> > >
> > > **Comment 2:** Moreover, I am still bothered by the fact that the main result relies on several approximations. I understand that they are necessary to obtain the main result, but I actually don't know what it means to maximize the approximation. What if the approximation is too crude? The result of the procedure could be meaningless.
> > >
> > > **Reply:** We make approximations in order to simplify the analysis so as to obtain solvable issues and we have discussed the estimation errors as well as validating the theorems in real scenarios.
> > >
> > > In order to validate such approximations, we designed extensive experiments. The experiment results are consistent with our designs. For example, to test whether our approximations are reasonable, we conduct experiments to verify whether we can use GSNR (Definition 4.1, which is a information model based on our theorems), to evaluate the quality of local datasets. Fig. 5 (b) shows that our model of GSNR can distinguish the information quality even the information is partitioned with Pareto distribution (the number of labels for different participants are displayed on the bottom of Fig. 5 (b)). Moreover, Fig. 6 shows that, if we maliciously manipulate local data, the model can recognize these changes by decreasing the GSNR dramatically.
> > > In summary, the real-world experiment results validate our approximations and our theoretical results are consistent with the experiment results, which demonstrate that our approximations are reasonable.
> > >
> > > **Comment 3:** Finally, I think that the exposition could be improved. The main message is not easy to understand. For example, in the current version, Thm 3.7 seems to be an easy math statement about the Wasserstein distance between Gaussians. I know that this is not the case, and that the Gaussians appearing here are related to the original SNR problem, but this is the kind of issues that the reader has to deal with.
> > >
> > > **Reply:** Thank you for your advice. We have rephrased the corresponding exposition in the revision.
> > >
> > > [1] Rudin, Cynthia, et al. *Interpretable machine learning: Fundamental principles and 10 grand challenges.* Statistics Surveys 16 (2022): 1-85.

---

> > > > ### Comment · Reviewer_8mk1 · 2022-11-18
> > > > **Thanks for the answer**
> > > >
> > > > OK, so I should treat this paper as a methodological/experimental paper. I will keep my score though, because I am not sure if the exp are significant enough for ICLR. And this I am not an expert on the experimental side, I will keep my low confidence level.

---

> ### Author Response · Authors · 2022-11-13
> **Author Reply**
>
> We thank the reviewer for detailed comments and reviews.
>
> **Comment 1:** GSNR is computed on the basis of several approximations. We don't know how far is the estimated optimal number of local updates from the true optimal number of local updates. In general, the fact that the approach is based on several approximations is really not discussed.
> The optimal numbers are obtained after making several approximations: CLT, Lemma 3.4. This is not recalled, nor discussed.
>
> **Reply:** The reason why we need to make approximations is that we need to decide the number of local updates before the local optimization process. Particularly, at the beginning of each communication round, we only have the information of initial parameters and the local data, and thus we need to make approximations to predict the optimal number of local steps with such limited information.\
> However, with such limited information, we can only make approximations to predict the optimal number of local steps. We provide upper bounds for the estimation errors. Specifically, Lemma 3.2 uses the CLT to approximate the gradient distribution. The estimation error is proportional to $ \frac{\sqrt[4]{rank(\Sigma)}} {\sqrt{B}}$, which means a larger batch size leads to a smaller estimation error. While for Lemma 3.4, the estimation error is $\mathcal{O}((n η_r)^2 LG)$ and a suitable relationship between $n$ and $η_r$ leads to a bounded estimation error. We have added Remark 3.3 and 3.6 to discuss estimation errors in the revision.
>
> **Comment 2:** The goal of the GSNR approach is to accelerate the training. Acceleration is only shown experimentally
>
> **Reply:** We can show the theoretical relationship between maximizing GSNR and the faster convergence based on Shannon's Formula, i.e., $C=W*log(1+\text{SNR})$ (see [Elements of Information Theory](http://staff.ustc.edu.cn/~cgong821/Wiley.Interscience.Elements.of.Information.Theory.Jul.2006.eBook-DDU.pdf) for details). Specifically, the channel capacity of information transmission $C$ is decided by GSNR, and a larger GSNR leads to a larger channel capacity for information transmission, which means the aggregation server gets richer information by the same communication rounds with a larger GSNR. With identical amount of information, the algorithm with larger GSNR requires less communication rounds, i.e., it achieves faster convergence.\
> Moreover, we extensively measure the convergence performance under different settings and results show that FedGNSR achieves $1.69\times$ speedup on average, which also validate our theoretical analysis. We clarifies this issue in the revision.
>
>
> **Comment 3:** Is Lemma 3.4 saying that running gradient descent is equivalent to running gradient descent without updating the gradient? This approximation seems too crude to give a good approximation of the optimal number of local steps. Besides, I would have expected the optimal number of local steps to be just one (since the communication constraint is not encoded in the GSNR)?
>
> **Reply:** Lemma 3.4 states that if we use the gradient calculated by initial parameters to estimate all gradients of local optimization process, the estimation error is $\mathcal{O}((n η_r)^2 LG)$. Moreover, if we consider learning rate decay, the estimation error decreases as the optimization process gets closer to the optimum.\
> Moreover, one local step is a choice for unbiased estimation, but it is not the optimal choice for accelerating training. Specifically, our analysis depends on the Information Theory, and based on Shannon's formula, GSNR decides the channel capacity of information transmission. A larger GSNR means a larger channel capacity for information transmission, which means for obtaining identical amount of information the server requires less communication rounds (i.e., it can achieve faster convergence). Hence the number of local steps which causes the maximum GSNR is the optimal value for acclerating training.
>
> **Comment 4:** In Eq. (5), $p_w$ are not defined. It is not clear what this equation means.
>
> **Reply:** Eq. (5) represents the wasserstein distance between the practical optimization path, i.e., $p_{\bar{w}}$, and the ideal optimization path, i.e., $p_{w_g}$. We have clarified the definition in the revision.\
> Specifically, $p_{w_g}$ represents the distribution of the gradients when we can gather all data ideally, and $p_{\bar{w}}$ denotes the practical distribution of gradients after aggregation.

---

### Official Review · Reviewer_kU4r · 2022-11-02

**Confidence:** 3
**Correctness:** 2
**Technical Novelty And Significance:** 2
**Empirical Novelty And Significance:** 3
**Recommendation:** 3

**Clarity, Quality, Novelty And Reproducibility:**

-

**Strength And Weaknesses:**

The paper provides a nice and (as far as I know) novel idea of how to use the statistical properties of local datasets in order to optimize the optimization procedure of the distribution problem.
However, I believe that the paper is too empirical and contains the following problems:

Major:
1. The paper does not provide convergence rate comparisons of the new method and FedAvg (see https://arxiv.org/pdf/1910.06378.pdf). It is not enough to say "The convergence of FedGSNR is obvious, as we just change the number of local updates." I believe that it is not a trivial task to obtain precise convergence rates for the new method.
2. Lemma 3.4 and the discussion implies that we can estimate $\eta_r \widehat{g}$ with $\eta_r \bar{g}$ if $\eta_r \approx 0.$ I think that it is not a strong result and probably misleading. This result virtually holds for any two random vectors, not only for gradients. In practice, we want to take $\eta_r$ as big as possible. Moreover, the smaller $\eta_r$, the worse we get the convergence rate.
3. In the proof of Theorem 3.5, the authors say: "the global gradient distribution can be approximately interpreted as normal distribution." It is a very weak argument for the main theorem of the paper. I would like to see more rigorous mathematical calculations.
4. Theoretically, it is not clear how the maximization GSNR is connected to faster convergence. For instance, is it possible to show that we can guarantee the convergence rate faster than the vanilla FedAvg if we maximize GSNR?

Minor:
1. In order to calculate $n^{opt},$ the method required the global $\mu_g$ and $\Sigma_g$ and local $\mu_l$ and $\Sigma_l.$ The authors propose to estimate them using sampling. It would be nice to have an alternative to Theorem 3.5.
2. How does the formula in (7) depends on $r$?
3. In order to estimate $\Sigma_g$ or $\Sigma_l$, would the method require a large batch size?



**Summary Of The Paper:**

In this paper, the authors investigate the possibility of optimizing a number of local steps in the FedAvg algorithm by minimizing the Wasserstein Distance between global and local gradient distributions.

**Summary Of The Review:**

While the idea of the paper is nice, the theory seems to be weak (see Strength And Weaknesses).

---

> ### Author Response · Authors · 2022-11-13
> **Author Reply**
>
> **Comment 4:** Theoretically, it is not clear how the maximization GSNR is connected to faster convergence. For instance, is it possible to show that we can guarantee the convergence rate faster than the vanilla FedAvg if we maximize GSNR?
>
> **Reply:** We can show the theoretical relationship between maximizing GSNR and the faster convergence based on Shannon's Formula, i.e., $C=W*log(1+\text{SNR})$. Specifically, the channel capacity of information transmission $C$ is decided by GSNR, and a larger GSNR leads to a larger channel capacity for information transmission, which means the aggregation server gets richer information by the same communication rounds. With identical amount of information, the algorithm with larger GSNR requires less communication rounds, i.e., it achieves faster convergence.
>
> **Comment 5:** In order to calculate $n^{opt}_1$, the method required the global $μ_g$ and $Σ_g$ and local $μ_l$ and $Σ_l$. The authors propose to estimate them using sampling. It would be nice to have an alternative to Theorem 3.5.
>
> **Reply:** The reason that we propose to estimate parameters by simapling is that the data distribution is unknown in practice and we only have some samples according to such an unknown distribution. To our best knowledge, the best method for the parameter estimation of model-free distribution is the Monte Carlo method (see <https://people.smp.uq.edu.au/DirkKroese/ps/whyMCM_fin.pdf> for details), which uses samples for estimation.
>
> **Comment 6:** How does the formula in (7) depends on $r$?
>
> **Reply:** The relationship between formula (7) and $r$ depends on the estimation error when we use $\hat{\mathbf{g}}$ to estimate $\bar{\mathbf{g}}$. In Lemma 3.4, we prove that $E|| \hat{\mathbf{g}}- \bar{\mathbf{g}} ||≤(n-1)n η_r LG$, where $n$, $L$, and $G$ are all constants, which implies if we condier learning rate decay, the estimation error converges in mean to 0. Moreover, convergence in mean implies convergence in probability, which is formula (7). To clarify this issue, we have added Remark 3.6 in the revision.
>
> **Comment 7:** In order to estimate $Σ_g$ or $Σ_l$, would the method require a large batch size?
>
> **Reply:** Yes, our method requires a large batch size $B$, leading to a small estimation error. Based on the Law of Large Numbers, the estimation error decreases as the batch size increases.

---

> > ### Comment · Reviewer_kU4r · 2022-11-13
> > **Respond**
> >
> > **Comment 1:**
> > It is not convincing. Can you provide a rigorous mathematical theorem that your algorithm convergences? I have many questions about this argument: What is $E_{\max}?$ $E_{\max}$ is probably also a random variable. Why is $E_{i,r}$ bounded and does not diverge with the number of iterations? In what sense do you get convergence? How does the randomness from your procedure influence convergence and why it preserves the convergence?
> >
> > **Comment 2:**
> > "*Particularly, the suitable learning rate leads to the better convergence performance...*"
> > In this comment, you are considering a nonsmooth convex problem (as an example, you take $y = |x|.$)
> > I am not aware of any results that show the convergence rates of FedAvg in this setting. Can you kindly provide it? If there is indeed some theory in this field, then the argument that we have to decrease the step size is convincing. But I would like to see proof of your method in this setting.
> >
> > Indeed, in the nonsmooth convex problems, it seems that we have to decrease a step size. But in the classical theory (see Introductory Lectures on Convex Optimization, Nesterov), the step sizes decrease as $O(\frac{1}{\sqrt{i}})$, while your step size decreases exponentially (see footnote 2).
> >
> > **Comment 3:**
> >
> > This is not very honest and fair...
> > You simply moved this argument to page 6:
> > "*On the one hand, according to Lemma 3.2 and 3.5, we can estimate the parameter distributions*"
> > The problem is still in the paper.
> > You still can not use Corollary 3.8 because the lemmas talk only about ***asymptotic convergence to the normal distribution.***
> >
> > **Comment 4:**
> > This argument is not convincing. The community and I want to see rigorous mathematical formulations and arguments. Can you state a theorem/property/lemma with proof (or only mathematical proof)?
> > How does the channel capacity is connected to the convergence speed? How do you define convergence speed? Why is this information useful for the server? How the channel capacity of information transmission is connected to the convergence FedAvg?

---

> > > ### Author Response · Authors · 2022-11-18
> > > **Thanks for your response**
> > >
> > > **Comment 4:** This argument is not convincing. The community and I want to see rigorous mathematical formulations and arguments. Can you state a theorem/property/lemma with proof (or only mathematical proof)? How does the channel capacity is connected to the convergence speed? How do you define convergence speed? Why is this information useful for the server? How the channel capacity of information transmission is connected to the convergence FedAvg?
> > >
> > > **Reply:** The rigorous mathematical relationship between the channel capacity and the convergence is really an important and difficult problem, which is correlated to the interpretability of information flow for Deep Learning. Moreover, the mathematical relationship between the information flow and the training of Deep Learning has been raised as a grand challenge [2] (please see https://arxiv.org/pdf/2103.11251.pdf for details). Our paper focuses on developing a new perspective to interpret Federated Learning through information theory and improve the performance by modeling the process as a problem of information transmission.
> > >
> > > [2] Rudin, Cynthia, et al. *Interpretable machine learning: Fundamental principles and 10 grand challenges.* Statistics Surveys 16 (2022): 1-85.

---

> > > ### Author Response · Authors · 2022-11-18
> > > **Thanks for your response**
> > >
> > > First of all, we thank the reviewer for the comments.
> > >
> > > **Comment 1:** It is not convincing. Can you provide a rigorous mathematical theorem that your algorithm convergences? I have many questions about this argument: What is $E_{max}$? $E_{max}$ is probably also a random variable. Why is bounded and does not diverge with the number of iterations? In what sense do you get convergence? How does the randomness from your procedure influence convergence and why it preserves the convergence?
> > >
> > > **Reply:** We have provided a mathematical proof for the convergence of FedGSNR with FedAvg based on the fact that $n_1^{opt}\leq B \sqrt{\frac{\mathbb{E}||g_g||^2}{\mathbb{E}||g_l||^2}}$ (please see Lemma E.1 and E.2 for details) in Appendix E.
> > >
> > > 1. $E_{max}$ denotes the maximum of $E_{k, r}$ for different participants and different communication rounds.
> > > 2. Lemma E.8 states that the divergence of $ w_k^t $ at each time t, i.e., $ ||w_k^t - \bar{w}^t ||^2 $, has an upper bound of $\frac{4η^2_t E^2_{const} BG^2 L}{\mu}$, which is independent of $E_{max}$. Particularly, $E_{const}$ is a constant decided before the training phase, which influences our procedure by $E_{k,r} = n_{1,k,r}^{opt}*E_{const}$.
> > > 3. Our algorithm is convergent when the loss function satisfies L-smooth and u-strong convexity.
> > > 4. Our procedure decides the number of local updates by $n_1^{opt}$. Moreover, $n_1^{opt}$ increases as $\mathbb{E}||g_l||^2$ decreases, which means the number of local steps increases when the norm of local gradient decreases, and the total influences of local optimization process is upper bounded. We theoretically prove that based on our procedure, $|| w_k^t - \bar{w}^t ||^2$ is upper bounded at each time t, and hence it preserves the convergence.
> > >
> > > **Comment 2:** "Particularly, the suitable learning rate leads to the better convergence performance..." In this comment, you are considering a non-smooth convex problem (as an example, you take ) I am not aware of any results that show the convergence rates of FedAvg in this setting. Can you kindly provide it? If there is indeed some theory in this field, then the argument that we have to decrease the step size is convincing. But I would like to see proof of your method in this setting.
> > > Indeed, in the non-smooth convex problems, it seems that we have to decrease a step size. But in the classical theory (see Introductory Lectures on Convex Optimization, Nesterov), the step sizes decrease as $\mathcal{O}(\frac{1}{\sqrt{i}})$ , while your step size decreases exponentially (see footnote 2).
> > >
> > > **Reply:** Li et al. [1] (please see https://openreview.net/forum?id=HJxNAnVtDS for details) find that when we set local steps $E>1$, FedAvg cannot converges to the optimum without learning rate decay (see Thm. 4 in [1]). Particularly, the solution will be $\Omega(η)$ away from the optimal. In our paper, Thm. 3.7 indicates that in non-iid scenario, without learning rate decay, the minimal Wasserstein distance between ideal optimization process and the practical optimization process will not be $0$, which means the gap between the practical method and the ideal optimization process will consistently exist when the ideal optimization process approaches to the optimum.
> > >
> > > Footnote 2 is not an essential part of our algorithm, it’s only an example of the existing learning rate decay method that has been widely used. Based on Lemma 3.5, the estimation error converges to 0 if the learning rate approaches 0. Thus, if we use the learning rate decay method as $\mathcal{O}(\frac{1}{\sqrt{i}})$, the estimation error also converges to 0.
> > >
> > > **Comment 3:** This is not very honest and fair... You simply moved this argument to page 6: "On the one hand, according to Lemma 3.2 and 3.5, we can estimate the parameter distributions" The problem is still in the paper. You still can not use Corollary 3.8 because the lemmas talk only about asymptotic convergence to the normal distribution.
> > >
> > > **Reply:** Corollary 3.8 proves that the optimal number of local steps to minimize the distance of two Gaussian distributions is $ m \cdot n_1^{opt} $. The theorem still holds without Lemma 3.2 and 3.5. However, Lemma 3.2 and 3.5 state that it's reasonable to estimate the practical model of gradient distributions, which is extremely complex, by Gaussian distributions. Hence, in practice, we can use $m \cdot  n_1^{opt}$ to estimate the optimal number of local steps for training. Our extensive experiment results validate that our approximations are reasonable.\
> > > We move the argument to page 6 because we need to discuss the approximations after Thm. 3.7.
> > >
> > > [1] Li, Xiang, et al. *On the Convergence of FedAvg on Non-IID Data.* International Conference on Learning Representations. 2019.

---

> ### Author Response · Authors · 2022-11-13
> **Author Reply**
>
> Thank the reviewer for detailed comments.
>
> **Comment 1:** The paper does not provide convergence rate comparisons of the new method and FedAvg (see <https://arxiv.org/pdf/1910.06378.pdf>). It is not enough to say "The convergence of FedGSNR is obvious, as we just change the number of local updates." I believe that it is not a trivial task to obtain precise convergence rates for the new method.
>
> **Reply:** We agree with the reviewer that obtaining precise convergence rate is not a trivial task. Here we mean that our FedGSNR must be able to converge. It does not necessarily indicate that the FedGSNR is significantly better than FedAvg with regard to convergence rate. Specifically, based on the convergence analysis of original method without FedGSNR, we can obtain an upper bound for the convergence rate of FedGSNR by substituting $E_{max}$ or $E_{min}$ for $E_{i, r}$, where $E_{i,r}$ represents the corresponding number of local steps and $1 \leq E_{min} \leq E_{i, r} \leq E_{max}$. Moreover, we conduct experiments to show that FedGSNR can always achieve better convergence performance. To clarify this issue, we have rephrased the description in the revision.\
> Our goal is to analyze federated leaning through a new perspective, i.e., the Information Theory, and we treat federated learning as a problem of information transmission. Based on Shannon's Formula, i.e., $C=W*log(1+\text{SNR})$ (see [*Elements of Information Theory*](http://staff.ustc.edu.cn/~cgong821/Wiley.Interscience.Elements.of.Information.Theory.Jul.2006.eBook-DDU.pdf) for details), we can enlarge the communication channel capacity $C$, which represents the capability of information transmission, via a larger Gradient Signal to Noise Ratio (GSNR). The larger channel capacity means that it requires less communication rounds for obtaining the identical amount of information. Moreover, the experiment results of $1.69\times$ speedup on average supports such a new perspective for analysis.\
> Moreover, the convergence rate is only one dimension of the convergence analysis. In non-convex scenarios (which are the general cases in machine learning), the convergence rate represents the rate to converge to a stationary point, and it cannot guarantee the global optimality.
>
> **Comment 2:** Lemma 3.4 and the discussion implies that we can estimate $η_r\hat{\mathbf{g}}$  with $η_r\bar{\mathbf{g}}$ if $η_r≈0$. I think that it is not a strong result and probably misleading. This result virtually holds for any two random vectors, not only for gradients. In practice, we want to take $η_r$ as big as possible. Moreover, the smaller $η_r$, the worse we get the convergence rate.
>
> **Reply:** We can use $\hat{\mathbf{g}}$ to estimate $\bar{\mathbf{g}}$ because the estimation error is upper bounded, i.e., $E|| \hat{\mathbf{g}}- \bar{\mathbf{g}} || \leq (n-1)n η_r LG$, where $L$, $G$, and $n$ are all constants. We can reduce the estimation error by chosing a suitable $η_r$. Moreover, the estimation error is $\mathcal{O}((n η_r)^2 LG)$ if we multiply $E|| \hat{\mathbf{g}}- \bar{\mathbf{g}} ||$ by $η_r$, i.e., $E||η_r \hat{\mathbf{g}} - η_r \bar{\mathbf{g}}||=\mathcal{O}((n η_r)^2 LG)$. Then if we consider learning rate decay, the estimation error decreases when the optimization process gets closer to the optimum. To clarify the issue, we have added Remark 3.6 in the revision.\
> Particularly, the suitable learning rate leads to the better convergence performance, and the learning rate decay is always an important part of convergence. For example, if we use SGD to minimize $f(x)=|x|$, a typical process is $x_{n+1}=x_{n}-η \  \text{sign}(x_n)$. Hence the learning rate should be $η \leq 2|x_{n}|$, otherwise, $x_{n+1}$ gets further away from the optimum 0. Actually, the algorithm only converges to $Ω(η)$ for any $η>0$, which means a large learning rate $η$ causes a large error of fluctuation. The corresponding conclusion for non-iid federated learning is proved by the existing work (see [*On The Convergence of FedAvg on Non-IID
> Data*](https://openreview.net/forum?id=HJxNAnVtDS) for details). We calrified this issue in the revision.
>
> **Comment 3:** In the proof of Theorem 3.5, the authors say: "the global gradient distribution can be approximately interpreted as normal distribution." It is a very weak argument for the main theorem of the paper. I would like to see more rigorous mathematical calculations.
>
> **Reply:** We provides a rigorous proof for Theorem 3.5 in the revision.\
> Specifically, Theorem 3.5 states that the minimal wasserstein distance of two Gaussian distribution denoted by $\mathcal{N}( η_r \mu_g,   \frac{η_r^2\Sigma_g}{B})$ and $\mathcal{N}( n η_r\mu_l,  \frac{n^2 η_r^2 \Sigma_l}{ B})$ with regard to $n$ is achieved when $n=n^{opt}_1$.

---

### Official Review · Reviewer_zXke · 2022-11-03

**Confidence:** 1
**Correctness:** 3
**Technical Novelty And Significance:** 3
**Empirical Novelty And Significance:** 3
**Recommendation:** 6

**Clarity, Quality, Novelty And Reproducibility:**

In Lemma 3.2: When the samples are significantly non-iid, we cannot apply the CLT in general. However, the authors apply the CLT to show Lemma 3.2. Could you explain why it is possible? In addition, I think that the statement should be "$\sqrt{B}(g - \mathbb{E}[g])$ converges to a multivariate normal distribution as $B\to \infty$,"  and the authors should clarify that what CLT theorem they used in the proof.

**Strength And Weaknesses:**

The proposed framework is general and seems to be applied to various existing methods. The non-iid data issue is important in this fields, and the authors address the problem well. Unfortunately, I am not familiar to this topic. Therefore, I cannot assess the exact strength and weakness of the proposed framework. I read the paper and could not find significant faults.

**Summary Of The Paper:**

This paper proposes a framework for federated learning with non-iid data. The authors tackle this problem by optimizing Gradient Signal to Noise Ratio (GSNR). They decompose local gradients calculated on the non-iid training data into the signal and noise components and then speed up the model convergence by maximizing GSNR. Based on this, they develop the FedGSNR method, which can be applied to existing gradient calculation algorithms.


**Summary Of The Review:**

The paper is well-written, although I could not evaluate it with confidence. However, there are several parts where theoretical explanation is insufficient. For example, the authors define the expected value in Section 2, but because the data is non-iid, we can consider samples that do not have the expectation. The authors more carefully discuss them because this paper discusses non-iid data. Furthermore, I could not understand why Lemma 3.2 holds. I would like the authors to clarify it.

---

> ### Author Response · Authors · 2022-11-13
> **Author Reply**
>
> We thank the reviewer for the constructive comments.
>
> **Comment 1:** In Lemma 3.2, when the samples are significantly non-iid, we cannot apply the CLT in general. However, the authors apply the CLT to show Lemma 3.2. Could you explain why it is possible? In addition, I think that the statement should be $\sqrt{B}(g−E[g])$ converges to a multivariate normal distribution as $B \to \infty$," and the authors should clarify that what CLT theorem they used in the proof.
>
> **Reply:** Classical CLT can be used to prove that for a specific participant, $\sqrt{B}(\mathbf{g}−E[\mathbf{g}])$ coverges to multivariate Gaussian distribution when batch size $B \to \infty$ because the local optimization process utilizes iid samples sampled from his own dataset. Regarding non-iid, it refers to the differences between the distributions of different participants. We only consider one participant when we prove the limiting behaviour of the local optimization process. Thus, we apply the classical CLT theorem to prove Lemma 3.2. We have clarified these issues in the revision.
>
> **Comment 2:** The authors define the expected value in Section 2, but because the data is non-iid, we can consider samples that do not have the expectation. The authors more carefully discuss them because this paper discusses non-iid data. Furthermore, I could not understand why Lemma 3.2 holds. I would like the authors to clarify it.
>
> **Reply:** The expectation exists in the non-iid scenario since the local optimization loss for each participant is the expectation of iid samples sampled from his own dataset, while the global optimization loss for non-iid scenario on the server is a convex combination of the local optimization loss.\
> Lemma 3.2 holds when we only consider one participant, where samples are iid sampled from a specific local dataset.

---

### Decision · Program_Chairs · 2023-01-20

**Decision:**

Reject

**Justification For Why Not Higher Score:**

Wrong and misleading mathematical statements

**Justification For Why Not Lower Score:**

N/A

**Metareview: Summary, Strengths And Weaknesses:**

The authors investigate the possibility of optimizing the # of local steps in FedAvg by minimizing the Wasserstein Distance between the global and local gradient distributions. The main idea of the work seems to be novel.

The work strings together a number of heuristic arguments presented as rigorous mathematical results. Some reviewers criticized the insufficient rigor, and I agree with this criticism. This alone is highly problematic, and is a reason for rejection. If the paper is of an empirical nature, it should be written as such, and the heuristic thinking that lead to the development of the method should have been clearly presented as such.

There are prior works that optimize  # of local steps in a mathematical rigorous manner (e.g., the ProxSkip method of Mishchenko et al, ICML 2022) which have not been discussed.

Some reviewers suggested borderline acceptance. However, both of these reviewers were not confident in his/her evaluation (confidence 1 and 2). I have read these reviews; and do not see sufficient reasoning/evidence to warrant acceptance.